# Aggregation-Induced Emission Properties in Fully π-Conjugated Polymers, Dendrimers, and Oligomers

**DOI:** 10.3390/polym13020213

**Published:** 2021-01-09

**Authors:** Antonio Sánchez-Ruiz, Ana Sousa-Herves, Juan Tolosa, Amparo Navarro, Joaquín C. García-Martínez

**Affiliations:** 1Facultad de Farmacia, Departamento de Química Inorgánica Orgánica y Bioquímica, Universidad de Castilla-La Mancha, C/José María Sánchez Ibáñez s/n, 02008 Albacete, Spain; Antonio.SanchezRuiz@uclm.es (A.S.-R.); Ana.Sousa@uclm.es (A.S.-H.); juan.tolosa@uclm.es (J.T.); 2Regional Center for Biomedical Research (CRIB), Universidad de Castilla-La Mancha, C/Almansa 13, 02008 Albacete, Spain; 3Department of Physical and Analytical Chemistry, Faculty of Experimental Sciences, Campus Las Lagunillas, Universidad de Jaén, 23071 Jaén, Spain; anavarro@ujaen.es

**Keywords:** Aggregation-Induced Emission, AIE, Aggregation Induced Emission Enhancement, AIEE, fully conjugated, π-conjugated, luminescence

## Abstract

Aggregation-Induced Emission (AIE) in organic molecules has recently attracted the attention of the scientific community because of their potential applications in different fields. Compared to small molecules, little attention has been paid to polymers and oligomers that exhibit AIE, despite having excellent properties such as high emission efficiency in aggregate and solid states, signal amplification effect, good processability and the availability of multiple functionalization sites. In addition to these features, if the molecular structure is fully conjugated, intramolecular electronic interactions between the composing chromophores may appear, thus giving rise to a wealth of new photophysical properties. In this review, we focus on selected fully conjugated oligomers, dendrimers and polymers, and briefly summarize their synthetic routes, fluorescence properties and potential applications. An exhaustive comparison between spectroscopic results in solution and aggregates or in solid state has been collected in almost all examples, and an opinion on the future direction of the field is briefly stated.

## 1. Introduction

George Gabriel Stokes, in 1852, coined the term “fluorescence” to describe the process of the emission of blue light produced by fluorspar ore when irradiated with ultraviolet light [1]. Since then, luminescence phenomena have attracted a great deal of attention in the scientific community [2]. Such is the importance that, in the last 40 years, nine Nobel Prizes in Chemistry or Physics have been directly related to the interaction between light and chemical compounds, mostly organic. Photosynthesis, conductive polymers, green fluorescent proteins, light-emitting diodes or optoelectronics would be some of these awarded discoveries [3]. Among all luminescent materials, organic compounds have the advantage of being easily prepared and tuned for specific properties. From a structural point of view, the organic molecules capable of both interacting with visible light and producing luminescence in their different forms, are constituted by a skeleton that contains π-type electrons or non-bonding orbitals. In addition, if double or triple bonds alternate with single bonds in the molecule, there is a delocalization of π-electrons from these orbitals throughout the structure known as conjugation [4]. If this conjugation extends throughout the molecular backbone, it is known as “fully conjugated” or “fully π-conjugated” systems.

A widespread geometrical parameter in conjugated systems is the Bond Length Alternation (BLA), defined by the average of the difference between the bond lengths of consecutive C–C single and double bonds. The determination of BLA allows us to estimate the degree of quinoidal or aromatic character in conjugated systems. HOMO-LUMO energy gap, usually referred to band gap, is another important parameter closely related to BLA, which strongly determines the intrinsic optoelectronic properties in conjugated systems [4,5]. Nevertheless, these properties do not depend exclusively on the conjugation of the π- and non-bonding electrons. The molecular scaffold together with the functional groups that decorate it, can have a great impact when tuning these properties, resulting in the design of elegant structures for challenging applications.

Among all the properties exhibited by conjugated compounds, luminescence is undoubtedly one of the most studied and still with interesting issues to be resolved. One of them is the control over the emitting properties in different environments, and that is closely related to maintaining these characteristics in the final device. For example, in the construction of organic light-emitting diodes (OLEDs), it is essential that the luminescent properties are maintained in the solid state, as luminescence can be change dramatically when going from solution to aggregated and/or solid states. Thus, the development of new emissive materials involves a rational design of the structure where not only intrinsic molecular properties must be considered, but also a wide range of intermolecular parameters that can be responsible of the mechanism involved in that phenomenon.

Despite the fact that dramatic changes in the fluorescence quantum yield (Φ) were described many years ago when going from solution to aggregated/solid states in early studies [6], in recent years the AIE (Aggregation Induced Emission), AIEE (Aggregation Induced Emission Enhancement), SLE (Solid-state Luminescence Enhancement) and ACQ (Aggregation Caused Quenching) acronyms have been widely used in literature to refer to a vast group of phenomena in which the luminescent properties are strongly affected by the molecular arrangement in solution and/or the packing in solid state. Although sometimes the acronyms AIE, AIEE or SLE are used indistinctly, the differences among them are well established. The AIE phenomenon refers to compounds that are not luminescent in solution but become emissive when aggregated or in solid state [7], AIEE refers to fluorescent compounds in solution with increased fluorescence when aggregated [8] or solid. SLE refers to compounds that are more fluorescent in solid state than in solution [9]. Parallel to the experimental observations, different theories and mechanisms have been proposed to decipher from a molecular and electronic point of view the origin of such intriguing phenomena [10]. In this respect, the theory of molecular excitons by Kasha et al. [11,12] has provided a successful description of the changes in luminescence after aggregation. According to it, luminescent properties depend on long range coulombic couplings due the interaction between molecular transition dipole moments of the neighbor molecules. Thus, the so-called J aggregates would produce bathochromic shift with respect to the non-aggregate state increasing or enhancing the emission, while the H aggregate would produce hypsochromic shifts and/or luminescence is suppressed. A more expanded Kasha’s theory of H- and J- molecular aggregates was exhaustively described by Spano et al. [13] considering the effects of vibronic coupling and intermolecular charge transfer. As a conclusion, short- and long-range excitonic coupling could interfere with each other to produce HH, HJ, JH and JJ aggregates resulting in a vast range of photophysical behaviors.

The restriction of intramolecular motions (RIM) mechanism is another model of more simplicity that has been successfully proposed to interpret the enhancement or the quenching of emission in many luminogens [14]. According to it, the absorbed energy by the molecules would be dissipated in solution through intramolecular vibrations (RIV) and rotations (RIR). However, when molecules are aggregated or in solid state, or the viscosity of the medium is increased, these molecular motions are blocked, and the molecules relax radiatively. A similar description is proposed when the luminescent properties are affected by the Z/E-photoisomerization through a C–C double bond. In solution, the photoisomerization is favored by the torsional motions of the molecule and, consequently, the emission is quenched. In contrast, when molecules become aggregated or the viscosity of the medium is increased, the photoisomerization is blocked resulting in bright states in solid state [15].

In addition to all these mechanisms, the presence of conical intersections between the potential energy surfaces of the ground and excited states has been also identified in many molecular systems to be responsible for non-radiative electronic relaxations. Thus, when the access to this intersection is restricted because of a consequence of significant geometry changes after excitation, the intersection is canceled, and the molecules relax radiatively. This effect has been called as Restricted Access to Conical Intersection (RACI) [10].

The interest aroused by this fascinating research field has led to the study and description of a many fluorescent chemical structures in the solid or aggregate state, as it is proven by the fact that in the last two years, at least forty reviews have been published related to this topic [16,17,18,19,20,21]. However, most of the molecular structures are discrete molecules or independent luminescent units within a non-conjugated skeleton. In the latter case, the ultimate properties of the integrated fluorescent moieties are, in most cases, very similar to those of the lead luminogens because the HOMO and LUMO molecular orbitals are essentially the same. On the contrary, when these luminogens are electronically connected through a conjugated structure, the effects produced on them are more important and the final photophysical properties can significantly change with respect to the initial luminogen. To the best of our knowledge, there is no review in the literature dedicated to the systems where luminescent units are repeated in a fully conjugated structure; thus, the hereby presented review, without being exhaustive, tries to account for fully conjugated oligomers, dendrimers, and polymers with the most outstanding AIE properties in recent years.

After this brief introduction where the characteristics and properties of the fully conjugated species have been described and the main processes that give rise to the emission of light in the solid or aggregate state have been summarized, we will proceed to recapitulate the most important synthetic methods for the preparation of conjugated compounds. Next, conjugated structures with repetitive units and AIE properties will be reviewed, starting with those prepared with a high structural control and very low polydispersity such as oligomers and dendrimers. If we consider oligomers as the repetition of a determined number of monomeric units in a perfectly known structure, two possibilities may be described according to how the monomers are included in the structure: (1) they are linked in a continuous form resulting in a more or less linear structure; (2) they are located around a central molecule or core to give star-shaped structures (Figure 1). Most of the totally conjugated oligomers described with AIE properties respond to this star-shape second form. In order to establish a selection criterion, only oligomeric structures with three or more repetitive units have been chosen. An extended version of these star-shaped molecules would be dendrimers, where monomers are sequentially placed in layers or generations around the core in a branched manner (Figure 1). Similarly, polymeric structures could be ordered depending if the luminogens are arranged in linear or branched out fashion (hyperbranched polymers). Within linear polymers, we find those whose monomeric units are ordered, although they contain more than one luminogen, and copolymers where several monomeric units are polymerized giving a random distribution of chromophores in the structure (Figure 1). Regarding the hyperbranched polymers, it is worth noting the recent interest in the so-called conjugated microporous polymers (CMPs) and covalent organic frameworks (COFs) and their properties. It seems logical that the next level of complexity would be the copolymers, in which several monomers are simultaneously reacted to give rise to a random distribution of them in new materials with new properties. Due to their structural complexity, it is difficult to make a correlation between their molecular structure and the final optical properties. In fact, in many of the cases, the studies of these new compounds focus more on evaluating their applied properties as materials than their fundamental optical properties and no spectroscopic data in solution or in solid state prior to their applications are described. The present review aims to collect the basic optical properties of fully conjugated structures, and for this reason copolymers are out of the scope of this work. Nevertheless, and in case some any reader is interested in investigating more about them, totally conjugated copolymers with AIE or AIEE properties have been described for applications in circularly polarized luminescence [22,23], electrochromic devices [24], electrochemiluminescence [25,26], photoacoustic imaging [27,28,29], photothermal therapy [30], sensing of superoxide ion [31] or drug delivery imaging [32]. Therefore, and along with this line, the most outstanding oligomers, dendrimers, and polymers with AIE properties are presented.

## 2. Synthetic Methodologies for π-Conjugated Structures

Although it is widely accepted by the scientific community that the first totally synthetic polymer was Bakelite, developed by Leo Hendrik Baekeland in 1907 [33], the truth is that, years before, in 1862, Henry Letheby described the synthesis of a fully-conjugated polymer: polyaniline, by electrochemical oxidation of aniline [34]. In the late 19th and early 20th centuries, the revolution in polymer chemistry resulted in the emergence of new polymers, methods of synthesis and various processes to modify them, which focused research on understanding these new structures, but left the characteristics of fully-conjugated polymers and their potential applications in oblivion. Hence, it was not until the late seventies that Heeger, Shirakawa and MacDiarmid developed a conductive polymer of polyacetylene doped with bromine and iodine vapors [35]. This discovery was the starting point in the development and search for innovative applications of this type of compounds. Such magnificent discovery was recognized with the awarding of the Nobel Prize in Chemistry to these three scientists in 2000 [3].

The different methodologies for the preparation of fully conjugated polymers can be classified into two: chemical polymerization and electrochemical polymerization [36]. In both cases, the precursor monomers are properly dissolved in the corresponding solvent. In the case of electrochemical polymerization, an electrolyte is also required to allow the circulation of the current. When a potential is applied, the polymer begins to deposit, forming a film on the surface of the working electrode. This method is especially indicated when the polymer is formed directly on the surface of the electronic device with that is going to be used because it produces very fine and homogeneous coatings. On the other hand, electrochemical polymerization is not indicated for mass production of polymer or to control the number of monomeric units that compose it as is the case with oligomers or dendrimers. In these cases, it is better to use chemical methodology for better results; therefore, most of the conjugated structures described in this review are prepared by chemical methodologies.

As mentioned before in the introduction, conjugated compounds alternate in their structure between simple and double or triple bonds. This means that the reactions that allow the coupling of monomers, either in a controlled way to prepare dendrimers or oligomers, or in a non-controlled manner to prepare polymers, must involve linking sp^2^ or sp carbons. Considering this, the most commonly employed synthetic methodologies for the formation of fully conjugated systems can be grouped into two types: coupling reactions or condensation reactions (Figure 2) [36].

Coupling reactions are catalyzed by metals that join sp^2^ or sp carbons through the formation of a single bond. Depending on the chemical characteristics of the molecules to be joined, a wide variety of coupling reactions have been described: Heck, Suzuki, Sonogashira, Stille, Negishi, etc. In most of them, palladium is the most frequently used metal. However, other metals such as nickel or copper are also employed in Kumada or Ullmann reactions, respectively.

On the other hand, condensation reactions normally involve the generation of a carbanion in a sp^3^ carbon that reacts with an electrophile. In this type of reaction, the resulting formed bond is a double bond from an elimination step in the condensation reaction. The most commonly used reactions of this type are, Wittig, Horner-Wadsworth-Emmons, Knoevenagel, McMurry, Wessling, or Glich (Figure 2). As an alternative to these two large groups, there are less employed reactions that allow obtaining conjugated systems, such as the Ziegler–Natta reaction, dehalogenation reaction, olefin metathesis reactions, or heat-induced reverse Diels–Alder reactions. The choice of the chemical reaction will ultimately depend on the structures to be prepared and the chemical characteristics of the precursors.

## 3. Fully Conjugated Oligomers

Since the concept of AIE was presented [7], a great number of molecules have been described with these properties, and the simplest ones have been used as luminogen scaffold that induce the emission in solid state. Among the most recurrent we can find tetraphenylethylene (**TPE**), triphenylamine (**TPA**), **BODIPY**, **pyrene**, **carbazole**, **fluorene**, or **phenothiazine** (Figure 3). Specifically, **TPE** has been widely used since it was first described by Tang et al. [37] This molecule can be considered as an oligomer where four benzene units repeat around a double bond. In solution of a good solvent such as acetonitrile or THF, the benzene rings of **TPE** show rotational and vibrational motions that interrupt the conjugation between them and it results in absorption of light in the ultraviolet region (λ_max_ = 299 nm). These intramolecular motions also produce a loss of energy through non-radiative relaxation processes and totally quench the fluorescence of this molecule in solution. In the solid state, however, these motions are blocked through a typical RIM mechanism, resulting in an increase in the fluorescence (λ_max_ = 475 nm) of up to 49% quantum yield [38]. Therefore, it is not surprising that it has been widely used as a benchmark core for the attachment of more complex conjugated structures. An exhaustive search yielded more than eighty different conjugated oligomers whose molecular center was the **TPE**. It is not the aim of this work to collect all of them, but we will present some of the most outstanding ones.

The compound that has demonstrated more uses and applications among the structures with a **TPE** center decorated with conjugated systems is undoubtedly the compound **1** (Figure 3). The optical properties of this compound are similar to those of **TPE**, with a fluorescence emission at 545 nm and a Φ = 30% in solid state [39]. The rigid structure and its four acid groups in the periphery make it an excellent candidate for the interaction with metals such as Pt [40,41], Cu [42], or Ca [43]. These interactions block vibrational and rotational motions resulting in an increased fluorescence. In this sense, and together with the interest shown in recent years for new materials such as Metal-Organic Frameworks (MOFs), MOFs based on compound **1** of Zr [39,44,45], Zn [46,47,48], Cd [47], Hf [44], or In [46] have been described.

Different luminogens have been attached to the periphery of **TPE** in order to increase the RIM process and the first option that comes to mind is the incorporation of more **TPE** or analogous units as in compounds **2** and **3** (Figure 4). The optical properties of compound **2** were initially described by Wang et al. in 2010 [49] and subsequently revised by Tang et al. in 2013 [50]. The synthetic methodologies in both works are different and that, together with the use of different solvents, may be the reason why there are discrepancies in the photophysical properties reported for the same compound. The synthesis procedure used by Wang’s group is a condensation reaction based on the formation of the benzhydryl anion by reaction of diphenylmethane with butyllithium. This anion reacts with a benzophenone derivative to generate **TPE** units in the periphery. In contrast, Tang et al. prepared compound **2** by a Suzuki coupling between the corresponding arylboronic acid and the tetrabrominated **TPE** core. Tang et al. studied compound **2** in 1,4-dioxane solution describing an absorption at 342 nm and a fluorescence quantum yield of 3.4%, which, although low, is higher than that described for **TPE**. On the contrary, Wang studied compounds **2** and **3** in THF describing an absorption of 331 and 328 nm and a quantum yield of 0.21 and 0.15%, respectively. Although there are slight discrepancies due to the use of different solvents, the slight redshift of the absorption in both cases compared to **TPE** suggests an increase in the degree of conjugation in these compounds. These differences are also observed in thin film fluorescence studies, where Tang’s group describes an emission at 510 nm with a Φ = 97% for compound **2** while Wang et al. report emission at 486 and 490 nm with quantum yields of 44 and 48%, for compounds **2** and **3,** respectively. Regardless of the numerical values reported, the compounds show an increase in solid state fluorescence compared to the solution in a magnitude equal to or even greater than **TPE**.

In the same work, Tang et al. [50] also described compounds **4** and **5** by identical synthetic procedure used for compound **2** and placing **TPA** or 9,9′-spirobi[fluorene] in the periphery of the **TPE** core (Figure 5). In a similar way that for compound **2**, the substitution of the peripheral **TPE** by these luminogens also results in compounds that show a very low luminescence in solution but very high in solid state (Figure 5). This indicates that the **TPE** in the center of the conjugated system conditions and determines the RIM processes that give rise to the AIE effect.

Other luminogens such as 7,10-diphenylfluoranthene (**6**) or **phenothiazine** (**7**) have also been incorporated into the periphery of a **TPE** core. Compound **6** was prepared by Diels–Alder cycloadition at 210 °C using **TPE** with four triple bonds in the periphery (1,1,2,2-tetrakis(4-ethynylphenyl)ethene) as dienophile (Figure 6, left) [51]. In THF solutions, compound **6** presents two absorption bands centered at λ = 304 and 377 nm, each of them attributable to the π-π* transitions of each of the fluorophores that form it. The solution emission results in a bluish-green light with a maximum at 525 nm and a Φ = 12%, which is significantly lower compared to the quantum yield of other fluoranthene derivatives (ca. 31%) [51]. Again, this suggests that **TPE** as a core extends the RIM effects to the peripheral groups. Although the quantum yield in film described is only 5.4%, an AIEE effect is observed when aggregation is induced by THF/H_2_O mixtures for this compound. Initially, the fluorescence intensity falls as the water content increase up to 50% in THF/water mixtures. Higher content of water results in blue shift of the emission wavelength to 493 nm and a significant increase in the intensity of fluorescence (Figure 6, left). Compound **6** has been used as a fluorescence turn-off sensor of nitroaromatic compounds.

In the same vein, Misra et al. described the preparation of compound **7** (Figure 6, right) that incorporates strong electron donor **phenothiazine** units by means of a Suzuki cross-coupling reaction [52]. This results in a donor-acceptor system as demonstrated by the high Stokes shift values (10747cm^−1^) and the bathochromism in the maximum wavelength emission when the solvent polarity increases (from 564 to 684 nm). The compound presents very low quantum yields in THF solution (0.28%) comparable with those described for **TPE**. A Density Functional Theory (DFT) computational study suggests that the four phenyls of **TPE** and the **phenothiazine** units are non-planar and orientate in different directions. When the molecule is excited with ultraviolet light, it acquires a flat conformation causing the extension in the conjugation and shifting the emission wavelength to red. Aggregation of compound **7** was induced by adding different water fractions to a solution of the compound in THF. Molar fractions of water below 50% show a very low fluorescence and this increases progressively until reaching its maximum at molar fractions of water of 70% showing a clear AIE effect (Figure 6, right).

The solid-state luminescent properties of the molecules strongly depend on the molecular arrangement and intermolecular interactions and, in the case of compounds with AIE, their effects are even more important. Sometimes it is possible to change the molecular packing mode by means of various external stimuli such as shearing, grinding or rubbing. These mechanical processes do not produce any chemical structural damage, but planarization of the twisted conformation is induced resulting in color changes of the absorbance or emission that can be reversible or irreversible. This is known as mechanochromism or mecanofluorochromism [53]. As the **TPE** core induces a torsion in the conjugated branches, this type of compounds seems to be good candidates to present this kind of properties. When compound **7** was mechanically ground in a mortar and pestle, the maximum emission wavelength shifted from 498 to 545 nm. Subsequently, the exposure of the pressed material to dichloromethane fumes resulted in a return of the emission to values of 504 nm.

Another example of mechanofluorochromism has been described by Son et al. for compound **8** [54]. This molecule was prepared by a sequential combination of the Suzuki and Knoevenagel reactions in order to introduce four α-cyanostyrene into the periphery of the **TPE** core. Similar to compound **7**, **8** presents AIE effect when the aggregation is induced with THF/H_2_O mixtures reaching Φ = 73% (water content of 80%). When compound **8** was slightly ground, the maximum light emission suffered a dramatic red shift of 27 nm, going from 525 to 552 nm. This process is perfectly reversible by exposing the ground solid to dichloromethane vapors which cause the maximum emission to revert to 525 nm again (Figure 7). The authors present XDR or DSC studies revealing that during the grinding process the crystalline structure of the solid and the interactions between the molecules were destroyed to become an amorphous phase that can be recovered with dichloromethane vapors. When a piece of filter paper impregnated with a solution of this compound was wounded with a spatula to write “AIE”, the letters appeared in orange on a green background under UV light. The exposure of this paper to CH_2_Cl_2_ vapors for 2 min causes the erasure of the letters and only the green background is seen (Figure 7).

**Fluorene** is another chromophore that has been used to functionalize the periphery of **TPE** [55]. Specifically, compounds **9**, **10** and **11** (Figure 8) were prepared by means of Suzuki coupling conditions using Pd(PPh_3_)_4_ as catalyst and tetrabromo-**TPE** and boronic acid of the corresponding **fluorene** as reactants. The absorption spectra in dichloromethane solution for the three compounds are basically a contribution of each of the chromophores that constitute the molecule (Figure 8a). For example, compound **9** has a maximum absorbance at 329 nm attributable to the fluorenyl-phenyl units and a shoulder at 280 nm which is very close to the absorption of **TPE**. When the number of **fluorene** units is increased (**10**), the maximum absorption is shifted to red (373 nm) as a consequence of the increase in conjugation and the intensity of **TPE** absorption decreases as the fluorophore ratio changes. Something similar occurs when **TPA** is attached to peripheral **fluorene** units (**11**), the absorption maximum is shifted to red (368 nm) but **TPE** absorption at 306 nm is still appreciable in the spectrum. The photoluminescence in solution shows the same situation, the emission wavelengths depend on the luminescent units and the number that constitutes the molecule. In this way, compound **9** presents an emission maximum at 409 nm with a less intense peak at 531 nm, while for compounds **10** and **11**, the most intense peak corresponds to the one with the least energy (longest wavelength): 528 nm and 523 nm, respectively, leaving the peak with the most energy relegated to a lower intensity. The authors do not report the values of quantum yield in solution but they do mention that “*the emitting properties are poor due to the nonradiative decay associated to the rotation of the arms around the ethene core*” [55]. The absorption of these compounds in solid state and film is comparable with the results obtained in solution, apart from the fact that an increase in the absorption band at lower energies is observed (Figure 8b). In addition, a considerable hypsochromic shift of the emission maxima is observed when dissolution and solid state are compared. The authors attribute this effect to a steric hindrance that avoids the structural relaxation of the excited state which leads to a hypochromic displacement of the emission. The powder quantum yields for these compounds are high and correspond to values of 29, 77 and 84% for **9**, **10** and **11**, respectively. Taking advantage of the photophysical properties of compound **9**, an all-organic distributed feedback laser was fabricated as the gain material and exhibited an average threshold energy fluence of 60 ± 6 μJ/cm^2^ and emission in the green region.

Likewise, other molecules have been used as a core for the preparation of oligomers with AIE properties, such as **pyrene**, **BODIPY** (Figure 2) or, simply, benzene. Different derivatives of 1,3,5-tris(styrylbenzene) and 1,2,4,5-tetra(styryl)benzene with phenyl rings in different positions have been described for the design of propeller-shaped or butterfly-shaped molecules such as compounds **12** [15,56,57,58], **13** [59], **14** [60], **15** [61], **16** [15,57] and **17** [59] (Figure 9). These compounds were prepared by a condensation reaction of the corresponding aldehydes or ketones, except **15,** which was synthesized by cyclotrimerization of the **TPE**-acetylene in the presence of TaCl_5_-Ph_4_Sn as catalyst. In the case of compounds **12**, **13**, **16** and **17,** the nucleophile was generated from the phosphonate and condensed with the corresponding aldehydes or ketones by a Horner–Wadsworth–Emmons reaction with potassium tert-butoxide (Figure 9). In contrast, compound **14** was obtained from 1,3,5-tris(benzoyl)benzene by reaction with diphenylmethyl lithium (Figure 9).

A summary of the optical properties of these compounds is given in Table 1. In THF solution, the absorption values are very similar for all 1,3,5-trisubstituted benzene molecules and a slight red shift for compound **12** is observed when R = OC_6_H_13_ due to the electron-donor character of the substituent. The absorbance of 1,2,4,5-tetrasubstituted is shifted to a larger wavelength because *meta* conjugation is less effective than *ortho* or *para*, and a reduction of the HOMO-LUMO gap produces a bathochromic displacement of the absorbance bands. Concerning the data of the emission, a bathochromic shift is observed as the phenyl or stilbene units are added as an indication that the conjugation increases. The fluorescence quantum yield clearly decreases when the phenyl rotors are incorporated and moderate values are observed for compounds **12** and **16**, with just one phenyl-rotor per arm. Reorganization energy studies and Huang-Rhys factor calculations for the compound **13** and **17** [59] suggest that the non-radiative deactivation of this type of compounds is through a RIM mechanism similar than that of the **TPE** molecule.

When these molecules aggregate, red shifts in the absorption and emission bands are again observed compared to those obtained in solution and the fluorescence quantum yields are between moderate and high, except for compound **13**. Exhaustive DFT studies indicate that, although the enhanced fluorescence effect can be mainly attributed to a RIM mechanism, the *cis-trans* photoisomerization, excitonic coupling, or RACI processes must also be considered in such a way that the emission of these compounds in solid state will be determined by the greater or lesser contribution of each of these phenomena. On the other hand, it is possible to induce the aggregation of these systems, not only through solvent mixtures but also through the use of surfactants [58]. Thus, 1,3,5-tris(styryl)benzene compounds similar to compound **12** have been encapsulated in cetyl trimethyl ammonium bromide (CTAB) micelles giving rise to worm-shaped aggregates of a few hundred nanometers in size (Figure 10). The photographs of transmission electron microscopy (TEM) show lattices separation of 0.33 nm that corresponds to π-π interactions between the conjugated units and that is perpendicular to the longitudinal axis of the worm agglomeration suggesting a columnar-type aggregation within the micelle. The quantum yields of the compounds increase when they are encapsulated going from 28% in THF to 45% inside the micelle. This value is slightly higher (55%) when the micellar solution is lyophilized to bring the suspension to a solid state.

**Pyrene** derivatives are highly blue-fluorescent molecules in solution that have found widespread application [62]. However, they have been barely employed as solid state emitters because their planar conjugated structure experiences ACQ after excitation due to the π-π interactions between excimers [63]. If **pyrene** derivatives are expected to work as blue emitters in OLEDs, they must circumvent the issue of ACQ. Thus, the introduction of bulky groups at the periphery of the π-conjugated system was employed by Wang et al. to avoid excimeric interaction. They synthesized triphenylsilylphenyl and triphenylsilylphenyl(ethynyl) substituted **pyrene** derivatives **18** and **19** (Figure 11) via Suzuki and Sonogashira cross-coupling reactions with 1,3,6,8-tetrabromopyrene [64]. Fluorescence emission quantum yield of **18** was significantly higher than for **19**. DFT geometry optimizations showed that, despite the bulky groups, the longer distance between core and periphery in **19** allows the **pyrene** moieties to interact in an orthogonal orientation causing the partial quenching of the fluorescence. In any case, these structures do not strictly present AIE and, despite the promising optical properties in the solid state of **18**, no further experiments were presented to evaluate its possible application as OLED.

**Pyrene** cored oligomers have also been decorated with **TPE** units. Compounds **20**, **21** and **22** (Figure 12) work as good solid emitters and they undergo through AIE or AIEE processes depending on whether the fluorescence of the core is preserved in solution. All of them were prepared following standard Suzuki conditions for the coupling reaction of the corresponding boranes with 1,3,6,8-tetrabromopyrene and were tested in electroluminescent devices. Tang’s group synthesized oligomer **20** [65], while **21** and **22** were reported afterwards by Li’s group [66]. Compound **20** is stable up to 200 °C and in THF shows maxima at 398 and 462 nm for absorption and emission, respectively, and reported a Φ = 9.5%. Mixtures of THF/water above *f_w_* 60% shows AIEE effect as well as in solid film, where a bathochromic shift of the emission towards 483 nm and Φ = 70% is reported. The aggregate formation activates the RIM process which blocks the non-radiative channel and hence increases the light emission. Authors also reported that **20** exhibits an excellent performance as OLED with external quantum efficiency and current efficiency up to 4.95% and 12.3 cd·A^−1^, respectively. Compared to **20**, **21** and **22** present slight modifications: **21** is an isomer that connects the **pyrene** unit in *meta* to the **TPE** and **22** includes methyl groups to increase the torsion angle between the **pyrene** core and the **TPE** peripheries, causing a blue-shift in the emission in film for both compounds (443 nm and 453 nm, respectively). Again, both structures exhibit rise of the emission when water is added to THF solutions of them. **21** shows AIEE since the *meta* connection partially isolates the **pyrene** moiety and the compound is already fluorescent in THF, whereas **22** shows AIE since the vibration of the **TPE** units deactivates the inherent fluorescence of the core. Regarding electroluminescence, **21** exhibits a better performance indicating that *meta* linkage is more effective than the introduction of substituents to induce twisted conformations. However, both modifications do not result in any improvement over the performance of **20**.

**Pyrenes** have also been employed to decorate **TPE** instead of being located at the core of the structure [67]. In this case, the Suzuki coupling between a boronic acid derivative of 2,7-ditertbutylpyrene and 1,1,2,2-tetrakis(4-bromophenyl)ethene furnished oligomer **23**. Authors reported that it presents negligible emission in dilute THF solution, but when water is added (40 < *f_w_* < 80), enhanced luminescence intensity is observed with a deep blue emission peaked at 446 nm due to aggregation. Non-doped OLEDs of **23** achieved moderate electroluminescent performance with current efficiency of 2.54 cd·A^−1^. Nevertheless, doped devices employing up to 50% of doping concentration as the emitting layer doubles the current efficiency, which is among the best electroluminescence performance for solution-processed OLEDs based on blue luminogens [68,69].

Similar to **pyrene**, fluorophores based on **BODIPY**s have great luminescent properties in solution but suffer from ACQ in the solid state. To prevent this effect, AIE-active moieties such as **TPE** and **TPA** were installed in the **BODIPY** structure (Figure 13) [70,71,72]. **TPE** and **TPA** units were incorporated to **BODIPY** core through the Suzuki coupling reaction between bis-halide-**BODIPY** and the **TPE**-boronic or **TPA**-boronic acid to yield compounds **24**–**28** (Figure 13). When **TPE** is incorporated, the absorptions in THF solution for these compounds correspond to the absorptions of the chromophores that constitute them, the **TPE** units and the **BODIPY** core, resulting in a spectrum with two clearly differentiated bands at 310 and 536 nm. The emission in solution of these compounds comes from the reddest luminogen, i.e., **BODIPY**, with a fluorescence maximum between 567 and 689 nm and between moderate and high quantum yields (59% for **24**, 47% for **25**, 29% for **26**, 44% for **27**). Fluorescence intensity in THF solution at different wavelengths for these compounds showed that, upon addition of water, dynamic quenching occurred, as expected from the increase in polarity of the medium. When *f_w_* was increased over 80%, aggregation started to occur, and the intensity of the fluorescence began to increase reaching up to 50% more intensity than when dissolved in THF alone [70,71].

Alternatively to **TPE, TPA** units have been added to the periphery of a **BODIPY** (**28**) [72]. In this case, the optical properties in THF solution of compound **28** are very similar to **BODIPY** with maximum absorption at 540 nm, maximum emission at 641 nm and very low quantum yield (3%) despite having a very fluorescent unit such as **BODIPY**. As with the **TPE**-substituted **BODIPY**s, compound **28** shows an AIEE effect in THF/water mixtures. As the solvent mixture is enriched with water, the fluorescence intensity decreases to a minimum at *f_w_* = 60% from which point the addition occurs and the fluorescence intensity increases up to 30% from the pure THF value.

**TPA** is an excellent electron-donating group with many advantageous characteristics such as good solubility, photoluminescence and electrochromic behavior; however, as with **pyrene** and **BODIPY**, it cannot be used in practical applications due to its ACQ effect [73]. To address this issue, the group of Tang reported a couple examples of **TPA** incorporated as core decorated with three or four **TPE** units (**29** and **30,**
Figure 14) [73]. The compounds were synthesized by Suzuki reaction with boronic acid-**TPE** and the corresponding halogenated **TPA** derivative. Both compounds showed absorptions around 375 nm in THF solution, similar to those attributed for **TPA** units and Φ values below 1%. The progressive addition of water leads to an increase in fluorescence from *f_w_* = 40%, reaching a maximum for *f_w_* = 90% up to 97-fold of the value in THF (*f_w_* = 0%). This AIE effect also extends to the solid state and thin film, with both compounds reaching Φ of 91.6 and 100% for **29** and **30**, respectively. Alternatively, **TPA** units were also placed in the periphery of luminogens such as benzothiadiazole (**31**) [74] or tetraazaanthracene (**32** and **33**) [75] looking for changing **TPA** properties from ACQ to AIE (Figure 14). For instance, ultrabright red luminogens such as **31** are synthesized by Buchwald-Hartwig cross-coupling reaction between a dibromo-benzothiadiazole intermediate and **TPE** aryl amines in the presence of Pd catalyst [74]. In this case, the arylamino unit works as a donor and benzothiadiazole as the electron acceptor. Compound **31** is a red-emissive molecule with insignificant solution emission but efficient aggregate-state photoluminescence. Its Φ in solution and solid state were quantitatively measured, showing 0.35 and 34.1% values, respectively. DFT calculations were performed indicating that the HOMO levels of **31** were dominated by the orbitals from the benzothiadiazole core and arylamino units. However, the electron clouds of the LUMO levels were mainly located on the core and the adjacent phenyl rings. Such electronic distributions impart an intrinsic intramolecular charge transfer effect. Compound **31** adopts non-planar conformations with different twisting angles between different aromatic rings or double bonds. Such a molecular structure disfavors close π-π stacking interactions in the aggregated state, which enables it to emit efficiently in the condensed phase. Using a simple nanoprecipitation method, AIE dots from **31** were prepared and further used as efficient bioprobes. The fluorescent dots were employed for in vivo two-photon deep-tissue imaging of the blood vessels in vivo, allowing the accurate determination of the capillary diameters in mouse ears (Figure 14).

Tetraazaanthracenes decorated with **TPA** (**32** and **33**, Figure 14) were obtained via cyclocondensation reactions between 1,2-diaminoarenes and 1,2-diketones [75]. Experimental and theoretical studies demonstrated that attaching phenyl rotors to tetraazaanthracene facilitated the intramolecular rotations in solution that dissipate energy non-radiatively after excitation. Those intramolecular motions were suppressed upon aggregation to open up the radiative relaxation channel through a RIM mechanism, endowing these derivatives with AIE activity. As consequence, the Φ in THF solutions reported for compounds **32** and **33** are 19.7 and 1%, respectively. This indicates that the incorporation of a benzene ring leads to an increase in these rotational movements in **32** compared to **33**. Because of that, solid state of **33** shows an increase in quantum yield up to Φ = 8% and the maximum emission is displaced hypsochromically, while in compound **32** the shift is bathochromic, and the Φ drops to 4.9%. Finally, the covalent attachment of electron-donating **TPA** to the peripheries of the AIE core rendered red AIE luminogens with excellent photostability that could be formulated into nanoparticles and employed for long-term lysosome imaging (Figure 14).

As discussed in the introduction and seen throughout this section, most π-conjugated oligomers are star-shaped. Among the few examples that do not have this star-shape we can find oligomers of three **TPE** units joined linearly between the phenyls directly (**34**), through a double bond (**35**) or a triple bond (**36**) (Figure 15) [76]. These oligomers were prepared by coupling reactions of Suzuki, Wittig and Sonogashira, respectively. In THF solution the absorption maxima described are 341, 360, and 352 nm, clearly redshifted compared to **TPE** (299 nm) due to the increase in conjugation (Figure 15). The lower displacement is described for compound **34** which faces two phenyl rings, probably because a larger torsion between them reduces this conjugation if compared with compound **35** which joins the TPE units by means of more rigid and more conjugated double bonds. Emissions are also bathochromically displaced compared to TPE, showing values of λ_em_ = 493, 511, and 506 nm for **34**, **35**, and **36**, respectively. The Φ in THF solution is 1% and as water fractions are added to the THF solution, the aggregation of structures is induced and an increase in the quantum fluorescence yields is reached, reaching values of 90% for **35**, 76% for **35** and 74% for **36**.

Another example of linear oligomers consists of oligothiophenes with a **TPA** unit at one end. (Figure 16) [77]. The linear addition of the thiophenes on the oligomer results in widely tunable maximum emission wavelengths covering violet, blue, green, yellow, orange, red, deep red and NIR regions. The synthesis of the molecules are based on Suzuki coupling reaction of substituted 4-bromo-*N*,*N*-diphenylanilines with modified oligothiophen-2-ylboronic acids in the presence of a suitable Pd catalyst. XRD showed that the twisted conformation of the **TPA** segment extended the intermolecular distance between two parallel planes, reducing π-π intermolecular interactions and preventing emission quenching in its aggregation state. Moreover, the restriction of molecular motions (RIM) by intermolecular interactions enhanced the solid-state emission efficiency, and these **TPA**–oligothiophenes displayed quantum yields up to 40.79%. The emission colors were tuned by simple alteration of the HOMO–LUMO energy levels through the introduction of electron donor and electron acceptor substituents. Thus, increasing the lengths and strengths of electron-donor groups by the addition of thiophene units, or replacing the carbonyl group by dicyano groups, the emission wavelengths are dramatically shifted from 402 (violet) to 724 nm (NIR). These compounds could be employed as fluorescent probes for lipid droplet-specific cell imaging and cell fusion assessment, showing excellent image contrast and high photostability.

Other fully-conjugated oligomers with a non-star shape are the macrocycles **37** and **38** [78]. These macrocycles were synthesized through cross-coupling catalyzed reactions from the corresponding **TPE**-acetylene either Sonogashira for compound **37** or Eglinton for compound **38** (Figure 17). Upon irradiation in THF solution (at 355 nm for **37** and 365 nm for **38**), the fluorescence of the macrocycles was almost nonexistent, which was attributed to the rotation of the **TPE** phenyl rings around the alkyne axis. When the water fraction in THF was increased, aggregation began to occur and fluorescence intensity increased, with emission maxima located at 390 nm for **37** at a *f_w_* = 90%, and 530 nm for **38**, with the curve showing a continuously increasing trend with no maximum. Quantum yields in THF solution were 5 and 2% for compounds **37** and **38**, respectively. In 90:10 H_2_O:THF solution, these values were 92 and 80%. This AIE effect could be attributed to the restriction of the interconversion between the left- and right-handed helical conformations of the **TPE** moieties via the rotation of the phenyl rings, causing the locking of each **TPE** group in a single, non-convertible conformation that was observed in the X-ray structures. The aggregates in the solid state were found to be alternating trimers, although with random **TPE** helical conformations in each due to the absence of any sort of template.

The affinity of PEG cycles towards protons [79] was exploited in these compounds by employing chiral carboxylic acids as templates to induce the aggregation of the macrocycles in a definite helical conformation, an effect that could be observed through the circular dichroism (CD) spectrum. A mixture of macrocycles **37**/**38** and enantiomerically pure mandelic, chloromandelic or camphorsulfonic acids were studied in solution and in solid state. In solution, the only signals present in the CD spectra were those from the chiral acids, showing that no conformational induction was taking place. However, when deposited over a quartz surface and allowing to evaporate, strong CD signals were observed for macrocycle **37**, while **38** only exhibited weak CD signals when exposed to camphorsulfonic acid. This effect was attributed to the higher degree of flexibility of the macrocyclic structure in **38** when compared to **37**, which resulted in a fast rotation of the **TPE** phenyls in **38** and, therefore, to the quick interconversion between the different helical conformations of the **TPE** moieties. This chiral template effect was exerted only at the **TPE** moieties, not at the structure of the aggregates, as it was demonstrated by the FE-SEM images of the film itself. These results suggest that these materials could be employed in the design of chiroptical materials or chiral analyses when these compounds were bound to chiral molecules such as proteins or DNA. Other AIE macrocycles have also been described by Zhu et al. combining **TPE** and **TPA** in their structure [80].

In general, most of the fully conjugated oligomers with AIE properties present a star-shaped structure (although there are exceptions), which allows two effects: First, if the core is a AIE luminogen, like **TPE**, the effect that causes the non-radiative deactivation (RIM) is also extended to the branches and the effect is amplified. Second, the star shape allows us to place luminogen groups in the branches with steric hindrances that induce the non-radiative deactivation mechanisms via RIM because of rotational motions. In all cases, since they are fully conjugated molecules, aggregation leads to an increase in their degree of conjugation producing bathochromatic shifts at the maxima of absorption and emission. When looking at the solid or the agglomerated phase, rotational movements are blocked, and the RIM effect leads to an increase in fluorescence.

## 4. Fully Conjugated Dendrimers

Dendrimers are macromolecules with characteristics that put them halfway between oligomers and polymers [81,82]. Like oligomers, the dendrimers have a perfectly defined and known structure where the repetitive units are located around a nucleus or core and are repeated by layers called generations, resulting in a structure with a high degree of branching. Despite being monodispersed, they also have characteristics that remind us of polymers, such as high molecular sizes and weights, which translate into globular structures and a large number of functional groups. In addition, this globular structure creates holes or interstices that can also be used as hosts to encapsulate molecules or metallic nanoparticles [83]. The development and study of dendrimers with fluorescent properties began almost from the first time that these types of structures were discovered [84,85,86,87], and it is possible to find examples in which dendrimers have been used to increase the fluorescent properties of chromophores [88,89]. However, a few fully conjugated dendrimeric structures have been studied with the aim of evaluating whether they show effects of increased solid-state fluorescence or aggregation by themselves, and basically consists in the incorporation of luminogens in the dendritic backbone during the synthetic procedure.

There are two articles of dendrimeric architectures with **TPA**, both exploring the introduction of a multibranched triarylamine moiety not only to induce AIE, but to also evaluate those macromolecules as two photon absorption materials. Jiang et al. prepared two new AIE-active donor–π-bridge–acceptor (D–π–A)-type molecules containing a 1,3,5-triazine core as the acceptor and several **TPA** units to enrich the electron density of the system (Figure 18) [90]. Compounds **39** and **40** were prepared in a convergent way by combining the Ullmann and Horner–Wadsworth–Emmons coupling reactions. Both compounds show a negligible fluorescence in THF solution (Φ = 0.9%) while, when aggregation was induced by adding water fractions, Φ = 40% was achieved for *f_w_* = 90% and a blue-shift of the maximum emission wavelength. As it happened with monomers containing **TPA**, the torsional free motion in solution leads to the deactivation of the excitation by non-radiative decay. Compounds **39** and **40** exhibit very large two photon absorption cross-sections (4500–8629 GM). Authors also find that thin films prepared from them experiment an on–off fluorescence switching behavior that can be employed for the sensing of organic solvent vapors.

Tian’s group reported a series of conjugated dendrimers up to generation 2 (Figure 19) prepared following a convergent strategy which involves in iterative steps of Heck and Horner–Wadsworth–Emmons reactions [91]. They found a direct correlation between the increase in generations (Gn) and both the AIE effect in the nanoaggregates formed in THF/water mixtures and the quantum yield of films prepared with each dendrimer (Figure 18). As usual in conjugated dendrimers [87], these compounds have high molar extinction coefficients (10^5^ M^−1^·cm^−1^) and bands associated with charge-transfer processes. The maximum emission of these dendrimers is located above 565 nm and as generation increases, a band at lower wavelengths (~470 nm) arises due to the lower efficiency of the charge-transfer in certain areas of the dendrimer as the size of the molecule; additionally, a localized excitation of this part of the dendrimer is observed. The quantum yields in THF solution range from 1% for G0 to 7% for G2. The authors attribute this radiative loss to the phenomenon of twisted intramolecular charge transfer. When aggregation was induced by adding water fractions to the THF solution the Φ values increased to 85% for G2 in THF with a water fraction of *f_w_* = 80%. TEM photographs of this mixture of THF/H_2_O show aggregates whose size was estimated by DLS to be 103±5nm. In solid state, the quantum yields were also higher than in solution but lower than the aggregates in the THF/water mixtures. However, if a trend is observed in the three cases, solution, solvent mixtures and solid state in which increasing the generation of dendrimers, the quantum yields increase. This phenomenon is explained by a combination of intra- and intermolecular effects that cause rotational restrictions (RIR) and dipolar interactions with other molecules. The two-photon absorption of these dendrimers were measured by a two-photon-induced fluorescence method and the larger cross-sections were shown again by the second generation dendrimer (4400 GM in the nanoaggregates and 2072 in the films) proving the key role of the number of **TPA** units in the optical properties of the dendrimers.

Similar to **pyrene**, **fluorene** or **BODIPY**, **carbazole**-based luminescent materials are valuable candidates in the domain of photoluminescent and electronic devices, thus, it is a meaningful research topic to design and synthesize **carbazole**-based derivatives with AIE properties. **Carbazole** groups present good chemical stability and are also known for their excellent electron donating and charge transporting properties. Moreover, **carbazoles** can be easily functionalized at either nitrogen or aromatic backbones, facilitating their incorporation into different structures such as dendrimers (Figure 20).

As an early example, Chi et al. used a convergent approach to synthesize 2 generations of **carbazole** dendrimers: **42** (Figure 20). Employing a Horner–Wadsworth–Emmons reaction, monocarbazolyl and tricarbazolyl groups were attached to a distyrylanthracene core and the piezofluorochromic activity of the resulting dendrimers was greatly enhanced when compared to the original distyrylanthracene AIE behaviour [92]. In this case, and as described by the authors, the dendrimer **42** presents a strong fluorescence in THF solution (although Φ is not reported) that falls sharply when water fractions are added up to *f_w_* = 40%. From this point onwards, a red shift of the emission is observed and its intensity increases to values comparable to those from THF solution alone in the case of water-rich solutions. A later work determined that the fluorescence quantum yield in solid state of the lower generation dendrimer was 34.3% [93].

In a different work, Yamamoto et al. used a benzophenone core to synthesize three generations of **carbazole** dendrimers showing AIE and thermally activated delayed fluorescence (TADF) properties (**43**, Figure 20) [94]. Again, dendrimers were prepared in a convergent manner, while the coupling of the **carbazole** groups to the benzophenone core was performed through a variation of the classical Ullman reaction under mild conditions. In toluene solution, these dendrimers present an emission centered in 461 nm and with a Φ of 23.6 and 14.7% for the 2nd and 3rd generation, respectively. Similar to **TPA**-based dendrimer **41** described previously, this emission comes from a charge transfer process as suggested by the dependence of the maximum emission on the polarity of the solvent. When films were formed with them, an intense displacement of the emission band was observed at 493 nm and 500 nm with Φ that were 1.4 times higher than those observed in toluene solutions. This is reasonable because fluorescence from charge transfer states is sensitive to external environment and intermolecular interaction. In this case, the dendritic structure of **43** played an important role for both Thermally Activated Delayed Fluorescence and AIE activities in the neat films formed with a polymethyl methacrylate (PMMA) matrix; OLED devices were prepared using this material with fully solution processed organic multilayers and achieved a maximum external quantum efficiency of 5.7%.

**TPE** has also been used as the core for the preparation of **carbazole**-derived dendrimers (**44**, Figure 20). In a joint work, the groups of Ma, B. Li and X. Li constructed conjugated high-luminescence microporous polymer films based on these dendritic structures [95]. The dendrimers were synthesized through the Ullman coupling reaction, and their Φ in a THF/water solution was analyzed, showing a 110-fold enhancement for mixtures with high water content. Afterwards, microporous polymer films with an on–off fluorescence response were prepared and used as a sensing platform for volatile organic compounds (VOC). The detection mechanism was analyzed by theoretical calculations, which showed that electron-rich VOC vapors possess higher LUMO energy levels than the dendrimers. This facilitates the transfer of excited state electrons from electron-rich VOCs to the films, resulting in enhanced fluorescence intensity. At the same time, the VOC vapors can diffuse into the films through the microporous structures limiting the free rotation of **TPE** groups, thus reducing the nonradiative relaxation and increasing the fluorescence intensity. Therefore, the fluorescence enhancement of the films exposed to VOCs is achieved by the combination of AIE effect and electron transfer. Other AIE dendrimers have also been described combining **TPE** and **TPA** in their structure [80].

In the case of dendrimers, and as mentioned at the beginning of this section, since they are like the star-shaped oligomers but larger in size, the rotational movement effects leading to non-radiative deactivation and loss of fluorescence in solution are even greater. However, on the contrary, since the molecules are larger, localized emissions can be produced by parts of the dendrimers, which means that sometimes the fluorescence in solution is not negligible. Although in the solid or aggregate state the RIM effect is the most important in this type of structures, other intermolecular processes that increase the intensity of fluorescence in the solid or aggregate state cannot be ruled out.

## 5. Fully Conjugated Polymers

In this section, some examples of fully conjugated polymers with AIE or AIEE properties will be collected. Due to the great structural variety of the polymers, this section has been subdivided into four depending on the polymer structure and the number of luminogens. We will start with linear polymers with one luminogen, then we will describe linear polymers with two or more luminogens and then, some examples of hyperbranched polymers and covalent organic frameworks (COFs) will be presented. Finally, we will collect co-polymeric systems where two or more monomers have reacted to give rise to a random distribution of them along the structure. Due to the low homogeneity of the composition, it is very difficult to make a correlation between the fluorescent properties and the structure and, normally, what is pursued is the ultimate application of the prepared material. For this reason, in this last part, only the most outstanding copolymers will be collected, summarizing their main optical properties and the application for which they have been described.

### 5.1. Linear Polymers with One Luminogen

Likewise, oligomers and dendrimers, polymers with AIE properties are essentially constituted by luminogens such as those shown in Figure 3 and, once again, **TPE** leads the ranking as the most used structure. In this section, we will start describing different linear polymers containing **TPEs** joined through different conjugated connectors, which will be joined by linear polymers with other luminogens.

Linear polymers based exclusively on **TPE** structures were prepared through Suzuki cross-coupling reaction (Figure 21). Polymers **45** [96], **46** [96] and **47** [76] were obtained with an average molar masses (*M_w_*) of 15,200, 29,600, and 6100 and values of the polydispersity indexes (PDI) of 1.5, 1.7 and 1.73, respectively. The extension in the conjugation causes a redshift with respect to the **TPE** molecule of the absorption maxima of 21, 42, and 57 nm for **45**, **46** and **47**, respectively. This absorption is also shifted compared with similarly structured trimer **34,** indicating a higher degree of conjugation. All of them are practically non-fluorescent in THF and by adding small fractions of water, aggregation is induced, and the quantum fluorescence yield increases to 14.39%, 18.07% and 28% for a *f_w_* = 90%. Although an AIE phenomenon is observed, the quantum yields are significantly lower in aggregates than those observed for **TPE** or for oligomers with **TPE**, as previously described. Although non-radiative deactivation is produced by RIM, the authors point out that this lower Φ value may be due to structural defects in the polymer chains. These compounds have proven to be useful in the detection of nitro-explosive compounds. What is interesting is the compound **48** whose structure is very similar to that of the original compound **47** with the exception that two of the four phenyl rings of the **TPE** unit are replaced by naphthalene [97]. The maximum absorption for **48** (365 nm) is slightly red-shifted if compared to the compound **47** (356 nm) as a result of the inclusion of the naphthalene rings. The Φ in THF solution is 2.3%, very low, but somewhat better than for the linear **TPE** polymers mentioned. In a mixture of THF/water of *f_w_* = 90% the quantum yield of the aggregates reached the value of 30%. In this case, the rotational movements along the backbone of the polymers are very similar and the main differences lie in the naphthalene units that, in addition to being blocked in the rotational movements, produce better excitonic coupling with the conjugated polymer, resulting in better quantum yields. [97]

As an alternative to the direct bonding of **TPE** units for polymer formation, they can be linked by single conjugated components such as vinylenes, (**49**) [76], acetylenes (**50**) [76], phenylenes (**51**) [98] or phenylenacetylenes (**52**–**54**) [99] These polymers were synthesized by Wittig (**49**), Suzuki (**51**) or Sonogashira (**50**, **52**–**54**) reactions, respectively, and *M_w_* and PDI are summarized in Figure 22. In all cases, except for **51,** which, although the authors describe their synthesis, do not report any spectroscopic data, when the joint groups are incorporated, an increase in the conjugation is observed together with the corresponding red shift in the absorption wavelengths. The biggest displacement happens in compound **54**, since it combines electron-donating and electron-withdrawing groups in the structure that constitute D-π-A monomeric units with charge separation along the polymer. Quantum yield are low values in THF solution for all of them, increasing when aggregation is induced by the addition of water. As in the previous cases, a RIM effect is what explains these results. In the case of compound **54**, Φ of the aggregated compound is only slightly higher than that described in THF, as the presence of nitro groups gives rise to an intramolecular charge transfer process that leads to a non-radiative relaxation of the excited state, both in solution and in the aggregated state.

An interesting modification of a polymer very similar to the **52**–**54** compounds is the one designed by Bu et al. (**55**) [100], where crown ethers are placed along the backbone of the conjugated polymer (Figure 23). This modification hardly changes the optical properties of the polymer, with an absorption maximum at 378 nm and negligible fluorescence in THF solution are observed. However, when this polymer is combined with a bis(dibenzylamine) in an acidic medium, the crown ethers coordinate the ammonium groups, blocking the structure and preventing the rotational movements that deactivate the fluorescence. As a result, fluorescent and reversible vesicles depending on the value of pH can be formed (Figure 23).

Another elegant example is polymer **56** in which **TPE** units are bonded through furan rings that are formed in a one-pot of two tandem reactions (acyl-Sonogashira and Fiesselmann cyclocondensation) (Figure 24) [101]. An extensive study of the reaction conditions resulted in polymers with high molecular weights (up to 156,000 Da) and excellent reaction yields (93%). The optical properties are in line with those described so far, with maximum absorptions centered around 370 nm and negligible fluorescence quantum yields. The AIE effect begins to be observed as soon as water is added to the THF solution, reaching a maximum at *f_w_* = 80%. Analysis of the size of particles formed at the point of maximum fluorescence indicates a particle size of about 200 nm. The authors describe an application for this polymer as a turn-off sensor for Ru^3+^.

In addition to the interest that **TPE** linear polymers have elicited in the detection of nitro-explosive compounds, another interest has arisen in developing polymers with circularly polarized luminescence (CPL). CPL, the emission analogue of circular dichroism (CD), is the selective emission of left- or right-handed circularly polarized light originated from chiral chromophore systems. CPL is commonly used to investigate structural information on the excited state of chiral molecular structures with potential applications to generate optical devices and biologically active probes. These applications require fluorophores that are not only chiral but also provide strong fluorescence intensity in the aggregated state. So far, different CPL systems such as chiral organic conjugated molecules, polymers, or lanthanide complexes have been described. However, most of them suffer from ACQ effect in the solid state, which limits their applications. In this context, the AIE property provides a method to prepare CPL materials with high emission intensity in the solid state [102]. Most examples of CPL-active AIE-systems are based on chiral supramolecular structures where the AIEgen, typically **TPE**, is modified with chiral entities [103].

The first example of AIE-active polymer showing CPL was reported by Cheng et al. in 2013 (Figure 25) [104]. They synthesized a **TPE**-based chiral polymer (**57**) by the Sonogashira cross-coupling of **TPE**-diyne and a diiodo derivative of L-tyrosine. The generated luminogen showed AIE properties, with a weak fluorescence in THF solution (Φ = 0.4%) but become strongly luminescent as nanoparticle suspensions in *f_w_* = 95% (Φ = 3.6%). The resulting **57** can also exhibit a large CPL dissymmetry factor (g_lum_), a parameter normally used to evaluate the degree of CPL, in both solution and aggregate states.

Among CPL materials, chiral binaphthyl-based compounds are a family of especial relevance. Optically active 1,1′-binaphthol (BINOL) is one of the most important C2 symmetric compounds and it has been widely used in catalysis, fluorescence sensors or photonic devices. In the past few years, some of its derivatives combined with the **TPE** have been studied for their CPL properties [105,106,107,108,109]. In particular, the Cheng’s group first observed the aggregation-induced CPL of a TPE and binaphthyl-based conjugated polymer due to the formation of helical nanofibers in the aggregate state [105]. They synthesized four conjugated polymers incorporating chiral (*R*)-1,1′-binaphthyl moieties via Sonogashira or Suzuki reactions (**58**–**61**, Figure 26). All the polymers showed typical AIE phenomena and the relative emissive intensities could increase up to 160-fold when changing from THF to THF/water mixtures (Figure 26). None of the polymers had CPL properties in THF solutions, however, after aggregation in aqueous mixtures only the polymer linked with triple-bonds at the 3,3′-positions of binaphthyl moieties (**58**) exhibited obvious CPL. Further studies showed that this polymer self-assembled into helical nanofibers, which was a determinant parameter for the aggregation-induced CPL (Figure 26). These results remarked the relationship between the CPL intensities and the morphologies of the aggregates.

As we have seen, the incorporation of the TPE structure or similar in a polymer seems to induce the RIM effect to it and in all cases, polymers with AIE or AIEE properties are obtained. However, this does not happen with all luminogens; for instance, polymers based exclusively on **TPA** or fluorene lead to ACQ processes [110,111]. Sometimes, the simple addition of a disruptive element that breaks this sequence is enough to convert TPA polymers into luminescent ones when they are aggregated. Compounds **62** and **63** (Figure 27) are an example of this. These polymers incorporate furan to separate TPA units and were obtained by Stille cross-coupling reaction with high molecular masses and acceptable polydispersity [112]. The combination of an electron-rich polymer backbone with electron-withdrawing substitutes such as NO_2_ and ethylidenmalonitrile, leads to large Stokes displacements (6808 cm^−1^) and low fluorescence quantum yields in THF solution. In film, a slight red shift is observed in both absorption and emission maxima, indicating an increase in conjugation in addition to a strong increase in fluorescence quantum yield. Another option to transform ACQ polymers into AIEE polymers is the incorporation of TPE as a backbone decoration of the conjugated polymer. The poly(p-phenylenevinylenes) (PPV) are polymers widely studied for the development of OLEDs but present the disadvantages that in solid their fluorescence is quenched [113]. Different strategies have been designed to prevent this aggregation and to maintain the good emitting properties that they have in solution in the solid state [88]. Compound **64** is a PPV incorporating **TPE** hanging from the polymer backbone with *cis* stereochemistry for the double bonds. Prepared by ring opening metathesis polymerization (ROMP), which allows a high control in both the molecular weights and the PDI, the authors describe some polymers up to 12,350 Da and a PDI of 1.68. In THF solution, it shows an absorption maximum at 331 nm and an emission maximum at 511 nm with Φ = 45%. The authors also show how the polymer undergoes a *cis*-*trans* isomerization process when exposed to UV radiation (365 nm). The addition of water over the THF solution leads to an initial drop in fluorescence to a minimum of about 15% (*f_w_* = 50%) from which it begins to increase until it reaches a value similar to that of THF alone indicating a AIEE effect.

### 5.2. Linear Polymers with Two or more Luminogens

As we have just described, the development of polymer materials with a single luminogen not only makes more uncertain the possibility of presenting luminescent properties in a solid state, but also limits the possibility of tuning the optical properties and, therefore, its applications. For this reason, it is more frequent to find, in the literature, polymers that contain more than one luminogen in their structure. As we have commented in the previous section, polyfluorene suffers ACQ, and several authors have described polymeric structures in which fluorene is combined with TPE to invert this phenomenon (**65**) (Figure 28) [111,114,115,116]. In all cases the polymers were prepared by Suzuki coupling using dibrominated **TPE** and the boronic acid derivative of fluorene, obtaining molecular weights between 10,900 and 16,400 Da and PDI between 1.76 and 3.80. In THF, the compounds present a single absorption band centered at 366 nm, which indicates a conjugation between the two luminogen units, and a fluorescence emission maximum around 500 nm with Φ lower than 1%. When water fractions are added, the fluorescence increases until reaching Φ = 27% for a *f_w_* = 99%. These polymers have been used for the detection of explosives [111,114], lead ion sensors [116], and as efficient electroluminescent materials [114,115], With similar applications and prepared in the same way, TPE and carbazole polymers have also been described (**66**) [114,115]. Spectroscopic data are also comparable with a low THF emission centered at 508 nm and an AIE effect when aggregation is induced that can reach 28% for a *f_w_* = 90%. As with polymer **48**, **TPE** analogs with two phenyls replaced with naphthalenes, were combined with fluorene or carbazole to obtain polymers **67** and **68** [97]. When compounds **67** and **68** are compared with the previous ones, **65** and **66**, it is observed that, in addition to the emission band at 518 nm (associated with the di(naphthalen-2-yl)-1,2-diphenylethene unit), a band at wavelengths around 420 nm appears which, according to the authors, is attributed to the fluorene or carbazole units. This may indicate that the di(naphthalen-2-yl)-1,2-diphenylethene unit is more twisted and the conjugation between the luminogens is less efficient. The fluorescence quantum yields in THF solution are low, but somewhat higher than their counterparts with **TPE** (Φ = ~3%) and the fluorescence increases to Φ values of 25% and 33%, for **67** and **68** respectively when forming aggregates with THF/water mixtures (*f_w_* = 90%). A derivative analogous to compound **65** has been described as a mercury ion turn-on fluorescent sensor [117]. In a very similar manner, Wang et al. took advantage of the AIE properties of **TPE** and **TPA** to prepare multifunctional electrochromic materials (**69**–**72**) [118]. They synthesized a series of polymers by Stille coupling reaction in good yields, high molecular weights and good PDI (Figure 28). The resulting polymers were very soluble in *N*-methylpyrrolidone (NMP) and in other common solvents like THF in which all the spectroscopic characterization was described. Absorption spectra show a single band for each compound associated with the π-π* transitions of the conjugate polymer and the authors describe fluorescence emission values between 495 and 503 nm for a mixture of NMP: water of 1:9. In this mixture the reported Φ are 14.08, 69.11, 13.55 and 78.49% for **69**, **70**, **71** and **72**, respectively. The authors claim an AIE effect for these results.

Liu et al. first described the concept of polymerization-enhanced photosensitization, which means that conjugated polymers can show much higher photo-sensitization efficiency than their small-molecule counterparts. They synthesized a series of small molecules photosensitizers and showed that after polymerization, their efficiency in ^1^O_2_ production was significantly enhanced [119]. It was hypothesized that the numerous repeating units in the conjugated polymers would introduce more channels for intersystem crossing (ISC) and improve the light-harvesting ability. Another polymer (**73**) contained methoxy-substituted TPE as the electron donor and dicyanovinyl as the electron acceptor. Synthesized by Suzuki polymerization (*M_w_* = 17,000 Da, PDI = 1.45), the polymer was non-emissive in THF solution but showed an emission maximum at 640 nm in THF/water (*f_w_* = 99%) with a Φ of 6.2%. This polymer showed a very good effective ^1^O_2_ generation, which was 3.71-fold higher than that of the commercially available photosensitizer Ce6. In a later work, the same group used a similar structure to synthesize two conjugated polymer photosensitizers (**74** and **75**, Figure 29) for precise photodynamic therapy [120]. They adjusted the linkage position between the donor and acceptor moieties to improve the 2-photon absorption cross-section. The polymers were prepared through Suzuki coupling polymerization between **TPE** and dicyanovinyl-based monomers with average molecular weight and PDI described in Figure 29. The polymers were then nanoprecipitated rendering AIE dots with AIE properties. In particular, the Φ of the dots was measured in aqueous media and found to be 11.9% for **74**, and 3.1% for **75**. The AIE dots were employed for both in vitro cancer cell ablation and in vivo zebrafish liver tumor treatment via two-photon excited photodynamic therapy.

Other examples of linear polymers with two luminogens are shown in Figure 29. After **TPE**, **TPA** or **fluorene** are the most common luminogens in the search for AIE properties. Sometimes, polymerization reactions can result in very high molecular weight polymers and high polydispersity indexes which complicates their handling and use in certain applications. One strategy to control this is through dispersion polymerization. This type of polymerization plays with the solubilities of the monomer and polymer to control their size. For example, Suzuki’s reaction in geminal cross-coupling synthesis can result in polymers of molecular weights >70 kDa [80]. However, Zhu et al. have developed a methodology to prepare linear polymers that combines **fluorene** and **carbazole** in their structure (**76,**
Figure 29). This methodology is based on the fact that the monomer is very soluble in 1-propanol while the resulting polymer is non-soluble. Thus, obtained molecular weights are 13,100 Da with a PDI of 1.89. The polymer forms nanoparticles whose size can be controlled by the concentration of the initial reactant monomer and can range from 229 nm to 910 nm. These nanoparticles present fluorescent emission when irradiated at 400 nm in aqueous solution, showing a quantum yield value of 4.3%. This value falls dramatically when THF is added to the water suspension, which indicates that the aggregates are dissolved, and the molecules are dispersed until they are practically not fluorescent at a THF content of 99%.

In the above examples, we have seen how, by placing luminogens such as **TPE** units in the backbone of the linear polymer, it is possible to change the ACQ behavior to AIE. Alternatively, instead of being part of the backbone, **TPE** can be found decorating the fully conjugated polymer. Accordingly, we find examples of **TPA** (**77**, **78**, and **81**) [121,122], **carbazole** (**79** and **82**) [122,123] or **phenothiazine** (**80** and **83**) [124] polymers containing **TPE** or **TPE** analogs as substitutes along the polymer chain. All compounds in Figure 30 were synthesized though Yamamoto coupling reaction using Ni(COD)_2_ as catalyst with the *M_w_* and PDI shown in the chart. In all these compounds, no substantial differences are observed for the absorptions between the solution and the solid state. As **TPA**, **carbazole** and **phenothiazine** are electron-donating groups and **TPE** and **TPE** derivatives act as electron-withdrawing moieties in such a way that small D-π-A systems are formed in the polymer leading to red shifted absorptions and emissions compared to analogous homopolymers without **TPE**. In some cases, small shoulder bands at wavelengths around 300 nm are preserved as part of the units’ absorptions, but the one corresponding to the charge transfer process always dominates. The quantum yields in solution are generally low for all of them. A more extensive study of fluorescence in solution for compound **80** concluded that Φ was solvent dependent and could range from 2% for dioxane to 36.7% for DMF [124]. All the compounds show AIEE effect either in film or in aggregates in mixtures of solvents, typically THF/water or CHCl_3_/EtOH.

Finally, more than two luminogens can also be included in the structure of a linear polymer which means going one step further in the complexity of the systems and the combination of the properties of each of the luminogens. Some examples are shown in Figure 31. Compound **84**, composed of the luminogenes **TPE**, **TPA** and benzothiadiazole was prepared in an analogous way to compounds **73**–**75** and likewise presents polymerization-enhanced photosensitization [119]. As expected, the polymer showed typical AIE features in THF/water (*f_w_* = 99%) with a Φ of 5.5%. Another examples depicted in Figure 31 are the compounds **85** and **86** that mix in their structure **TPE**, **TPA** and **carbazole** (**85**) or **TPE**, **TPA** and **fluorene** (**86**) [125]. In THF solution, both compounds show very similar spectroscopic properties, with absorptions centered at around 383 nm, emissions at around 500 nm and quantum yields in the vicinity of 1%. In film, the emission maxima are shifted to red by about 8 nm and the Φ increase significantly up to about 30% in both cases. The authors use these polymers for the development of OLEDs and it is very interesting how, doping the films with 4,40-di(9H-carbazol-9-yl)-1,10-biphenyl, it is possible to increase the quantum yield of fluorescence in the doped film up to 63.3% for **85** and 56.1% for **86**. This results in emitting layers that show excellent results of maximum external quantum efficiency of 3.26% and current efficiency of 3.69 cd·A^−1^.

Undoubtedly, the polymers fully conjugated with AIE properties constitute the most abundant offer in the bibliography, forcing us to leave behind some of them in order not to make this review too long and monotonous [105,126,127,128,129]. In general terms, the presented selection gathers the most important aspects regarding the fully conjugated polymers and the AIE effect. The increase in the size of the molecules and the conjugated union between the luminogens make it progressively difficult to identify these luminogens independently in the linear polymers. It is more evident when there are more than one luminogen in the structure that leads to intramolecular electronic interactions between them, often resulting in D-π-A systems that displace the absorptions and emissions strongly to the red. Circularly polarized luminescence is an interesting property that opens a door to the development of compounds that interact in a chiral fashion with other molecules, especially biomolecules.

### 5.3. Hyperbranched Polymers, Conjugated Microporous Polymers (CMPs) and Covalent Organic Frameworks (COFs)

If the linear polymers are structures with the repetitive units joint by two points to lengthen the polymer chain, in the hyperbranched polymers, the repetitive units have more than two points of union that provides the polymer with ramifications and, consequently, different characteristics and properties compared to the linear polymers. These types of macromolecules have unique features, such as high density, low viscosity and many terminal functional groups that can be subsequently modified [130]. Among this type of highly branched materials, a special interest has recently emerged for two special kinds. One of them is composed of the so-called Covalent Organic Frameworks (COFs), which can be defined as highly ordered hyperbranched polymers with a crystalline and porous structure [131]; the second one integrates the group of Conjugated Microporous Polymers (CMPs), which are also porous materials but as opposed to COFs, exhibit a rather amorphous structure. Below are some examples of both, amorphous hyperbranched polymers, CMPs and COFs that have AIE or AIEE properties.

As has been seen throughout this review, the **TPE** molecule is always a reference in terms of employing it in the different molecular architectures and it is not different in the case of hyperbranched polymers and COFs. Using **TPE** as the main luminogen, Hu et al. [61] employed diacetylene-**TPE** as the monomer in a TaBr_5_-catalyzed photocyclotrimerization of the alkyne moieties to yield polymer **87** with very high molecular weight (Figure 32). This polymer was soluble in common organic solvents and was found to absorb at 335 nm in THF and weakly emit at 501 nm. Due to the polymeric nature of **87**, small quantities of water in the medium caused a significant rise in emission. The quantum yield in THF was 3.1% and 45% in THF:H_2_O with a *f_w_* = 90%. In solid state, however, Φ value was 47%, roughly the same as in the aggregated state in suspension. This result indicated that the polymeric structure greatly reduced the non-radiative energy loss, so there was not a big difference between the aggregated and solid-state fluorescence. This material was tested in the detection of picric acid as an example of explosives, which, upon interaction with **87** in THF/H_2_O mixtures, caused a steep fluorescence quenching that became more efficient with increasing picric acid concentrations. Complete quenching occurred at a picric acid concentration of 0.52 mM. Polymer **87** was also employed in the creation of photoresistant patterns via crosslinking the peripheric alkyne moieties and also had non-linear optical properties, among others [132].

CMPs such as the previously described, also find applications in light emitting and harvesting materials, due to the absence of aggregation that exists in standard conjugated polymers and that can give rise to the non-radiative dissipation of energy. An example of this kind of materials can be found in the work of Chen et al. who where **TPE**-based CMPs were prepared through either Suzuki or oxidative polymerization of different **TPE** monomers (Figure 33) [133]. These CMPs were tested in their nitrogen sorption and fluorescent properties in solid state. Compound **88** emitted in solid state at 530 nm due to the extended conjugation of the **TPE** moieties in the network, and the presence of thiophene in **89** red-shifted the emission to 590 nm. The authors describe this as an AIE effect although they do not report the characterization of the compounds in solution.

Using polymer **88** as one of the cores, Xu et al. [134] prepared core-shell nanoparticles composed of blue-emitting phenylene and yellow-emitting **TPE** polymers. The nanoparticles were prepared as described in Figure 34 (**90** and **91**). The outer shells defined the absorption of the materials, which red-shifted with the increasing thickness; this value was controlled through the molar ratio of starting monomers. Nanoparticles are fluorescent with a Φ value ranging between 15 and 32%.

The combination of **TPE** with other luminogens such as **fluorene**, **carbazole**, or phenyl, has allowed Li et al. [135] to describe a series of compounds (92–94, Figure 35) that, besides behaving as chemosensors of explosives, demonstrated that it is possible to implement them in OLED with good performance. Once again, the Suzuki reaction has allowed the coupling between the units yielding polymers with molecular weights between 3800 Da and 4800 Da and PDIs between 1.39–1.49. Spectroscopic data show that the polymers have similar absorption and emission profiles with maxima between 308 and 350 nm for absorption and 504–516 nm for emission. The fluorescence quantum yields in THF were practically nil for **92**, 1.1% for **93** and 5.2% for **94**. The addition of methanol led to aggregation, resulting in an increase in quantum yield values. The highest Φ value was obtained at 90% methanol content and presented for compound **94** which combines TPE with phenyl units (28.2%), then **fluorene** (23.8%), and finally **carbazole** with 17.6%.

A common feature of all these CMP materials is their high hydrophobicity that makes them completely insoluble in water and causes them to aggregate in organic solvents as long as there is water in them. Lee et al. [136] prepared a **TPE**-based CMP by a Sonogashira polymerization between tetra-(4-ethynylphenyl)-ethylene and 1,4-diiobenzene (**95**, Figure 36). By conducting this polymerization in the presence of polyvinyl pyrrolidone (PVP), the so obtained CMPs were found to have a bigger pore and nanoparticle size. As the PVP chains were intertwined with the nanoparticle, they imparted the material with enough polarity to make them easily dispersible in water. The absorption maximum was centered at 396 nm with an emission maximum at 545 nm in water with a quantum yield of 7.3%. These nanoparticles were evaluated as explosive sensors; several nitrophenol derivatives were tested and all of them caused quenching of the fluorescence to a minimum of a 60% with 0.1 mM concentration of nitrocompound.

By pairing **TPE** with a photoluminescent and good hole transporting compound such as carbazole, Wu et al. [137] prepared a hyperbranched polymer with AIE and electroluminescence properties (compound **96**, Figure 36). By means of Suzuki coupling, it is possible to obtain a polymer of 6800 Da and a PDI of 1.93. Optical properties showed a maximum in absorption at 296 and 341 nm, with an emission maximum centered at 512 nm and a quantum yield of 1.1% in THF. In the aggregated state in a THF:MeOH (1:9) solution, the emission experienced a slight blue shift to 508 nm, while a film of the material had a red-shifted emission to 517 nm. Quantum yield in the aggregated state was 29.6%, which represents a 27-fold increase over that from THF solution. Upon an electrical current application, a thin film of the polymer showed emission centered at 508 nm for a voltage of 5.2 V. As in the previous examples, the polymers could be employed in the detection of nitro compounds with a complete fluorescence quenching for a picric acid concentration of 30 ppm.

As mentioned in the above paragraphs, an evolution of CMP materials in this area is the synthesis of AIE-active COFs, which are crystalline porous polymers that integrate the organic units that compose them into periodic columnar π-arrays, thus creating an inherent porosity. This layering causes ACQ, which is prevented by incorporating AIE-active moieties into the structure. Dalapati et al. [138] created a phenyl-**TPE** COF (**97**) that exhibited an absorption centered at 390 nm and an emission maximum at 500 nm with a quantum yield in the solid state of 32 and 23% in cyclohexane and 28% in toluene (Figure 37). An interesting application of this material was the detection of ammonia, as the boronate moieties in the structure would readily form a Lewis acid-base complex with it. The k_q_ value of 1.4 × 10^14^ M^−1^ s^−1^ was reflected in the fact that 1 ppm of ammonia led to a 30% drop in the fluorescence intensity of the synthesized COF.

A refinement for these AIE-active COFs consists in incorporating chirality into the structure that would enable the resulting COF as a valid sensor for enantioselective analytical applications. Wu et al. [139] synthesized two COFs through imine formation polymerization between aniline-functionalized AIEgens and asymmetric BINOL-based dialdehydes (Figure 38). Compound **98** showed strong emission maximum at 380 nm in solid state, although excitation at different wavelengths yielded emission bands at 335, 420, 535, 576 and 630 nm, which showed the presence of different emissive species. To perform the analysis, nanosheets of (R)-**98** were suspended in acetonitrile and titrations were conducted. When exposed to α-pinene, emission at 380 nm decreased in both cases, but the rate of quenching caused by the (−)-enantiomer was much faster than that with the (+)-enantiomer, showing an enantioselective recognition together with a reduction in the Φ from 7.6 to 6.3%. If (R)-**98** was deposited in polyvinylidene difluoride membranes, a similar trend was observed when the membrane was exposed to α-pinene vapours. After 480 s of exposure, the membrane fluorescence had decreased in a 48% for the (−)-enantiomer and 8% for the (+)-enantiomer with a slight reduction in the Φ (from 1.5% to 1.1%) (Figure 38).

Regarding the fully conjugated hyperbranched polymers, it is one of the fields that have more projection at present. These structures combine concepts of great interest recently as are both the AIE effect and the CMP/COF materials. Many unknowns are yet to be elucidated, especially on how the structure affects the optical properties, if the interactions in a crystal are equal or different to an amorphous structure and how it affects the AIE properties, or if the generated pores can be played to enhance these properties.

## 6. Strength, Weakness, Opportunities and Threats in AIE Polymers

Although a full Strength, Weakness, Opportunities and Threats (SWOT) analysis would be very useful to reflect the current and future prospective of the AIE compounds in order to commercialize them, currently, the information available is very scarce and focused on fundamental research about their synthesis and photophysical characterization. Currently, only a very limited offer of TPE-derived molecules is marketed by Merck KGaA [140] or AIEgen BioTech Co. [141] but there is nothing concerning oligomers, dendrimers or polymers. Thus, its development is still in a very preliminary stage and not very oriented to an applied or commercial development. However, some ideas about Strengths, Weaknesses, Opportunities or Threats can be enumerated without being an exhaustive SWOT study.

*Strengths:* The applications of π-conjugated compounds cover areas ranging from the development of light-emitting devices, either lighting or display, solar cells, or sensors. The technological implementation of these compounds requires, in most cases, that they are in solid state in which polymers with AIE properties enhance their luminescent properties. The increasing interest expressed by the community in these compounds is evidenced by the progressive increase in the number of patents over the last 10 years (Figure 39, left), moreover, around 38% of the patents on AIE in 2020 were of polymeric materials. Probably, the advantages of polymers over other materials, such as versatility, low-cost of production, ease of manufacture, lightweight, portable, durable and more affordable products, make them good candidates for use in technological devices.

*Weaknesses:* However, despite these advantages, there are many questions still to be resolved about the control of AIE properties in macromolecular systems. As we have tried to collect in this review, most of the papers describe the luminescent properties of compounds but rarely try to provide a thorough explanation of why the phenomena occur, which complicates making correlations between molecular structure and photophysical properties. On the other hand, luminescent compounds are chemically constituted by highly hydrophobic elements. Since environmental sustainability has of great social interest worldwide, both the synthesis of compounds and their processability require the use of organic solvents, mostly pollutants such as THF, dichloromethane, ortho-dichlorobenzene, etc. This aspect is something that should be technologically solved in order to make the development of these materials more environmentally friendly.

*Opportunities:* Considering the above weaknesses, improvement opportunities could focus on two main areas: Search for new structures and search for new applications. The synthesis of new structures has other two clear objectives: to help to better understand how AIE processes occur in complex macromolecular systems, and to search for compounds that emit in less exploited regions of the emission spectrum or with low quantum yields. Figure 39 (on the right) shows the emission wavelengths and fluorescence quantum yields in solid state for all the compounds collected in this article as a representative sample. It can be seen how there is a great variety of compounds with different quantum yields between the blue and green regions in the spectrum. However, there are more limited compounds that emit with high quantum yields in purple, yellow, orange or red. As an alternative to these opportunities focused on the development of new molecular structures, the use of these compounds can also be extended to other areas such as biomedicine. Work has already begun in this area that focuses on the use of compounds with AIE properties for cell imaging, theranostics or phototherapy. The characteristics and requirements of the compounds for these applications are completely different from those used for the technologies mentioned above. This opens a wide field of research and development still to be explored.

*Threats:* Other types of materials with good luminescent properties are being developed in parallel and/or in competition with AIE polymers, such as quantum dots, graphene, borophene, phosphorene, transition metal dichalcogenides, perovskites, rare-earth-doped luminescent materials, among others (see references [142,143] and references therein). The great expectations placed on graphene have resulted in a disappointing eternal promise and borophene emerges as an alternative to overcome graphene [144]. The potential of both materials is very great, but also the disadvantages of processability and technology needed to reach the levels of polymers. On the other hand, quantum dots are more technologically developed, and it is possible to find electronic devices with this technology in the market. As a result of the development of quantum dots, AIE dots have been created to combine the advantages of quantum dots and AIE compounds in nanoscopic aggregates of AIE molecules.

## 7. Conclusions

In this review, we have tried to give a holistic vision about the development of fully conjugated structures in which luminogen units are repeated and combined to give compounds with AIE or AIEE properties, a perspective never treated in the bibliography. We have started with oligomeric structures whose level of molecular complexity is low and allows establishing easy correlations between the molecular structure and the final optical properties. Progressively, we have increased this degree of complexity to dendrimers, then to polymers with only one luminogen, polymers with more than one luminogen and finally to highly branched structures, such as CMP or COF. As has been shown, as the level of complexity of the structures increases, the constitutive luminogens lose their individual character because new intramolecular electronic interactions appear between the luminogens and the conjugated scaffolds, or even between the luminogens themselves, giving rise to more complicated systems from an electronic point of view allowing the construction of novel applied materials. Different luminogens have been shown throughout this work and, clearly, TPE is the mainstay as the directing agent for the induction of the RIM phenomenon as the main responsible for the AIE and AIEE effects. Although the synthetic reactions and mechanisms involved in those effects are well explored for conjugated molecules and macromolecules, there is still much room for improvement in the development of new fully conjugated structures with AIE and AIEE properties. In this sense, the study of the role of other phenomena alternative to RIM, such as *cis-trans* photoisomerization, RACI or intermolecular interactions in conjugated polymers, could help in finding new compounds with challenging applications. On the other hand, new AIE-active materials such as CMP, COFs or self-assembling compounds still need further exploration in the development in this field. In the case of the latter systems, aggregation is produced by an adequate and intelligent functionalization of the AIE molecule rather than by adding a mixture of solvents, a feature that would open the use of these compounds in fields such as biomedicine.

## Figures and Tables

**Figure 1 polymers-13-00213-f001:**
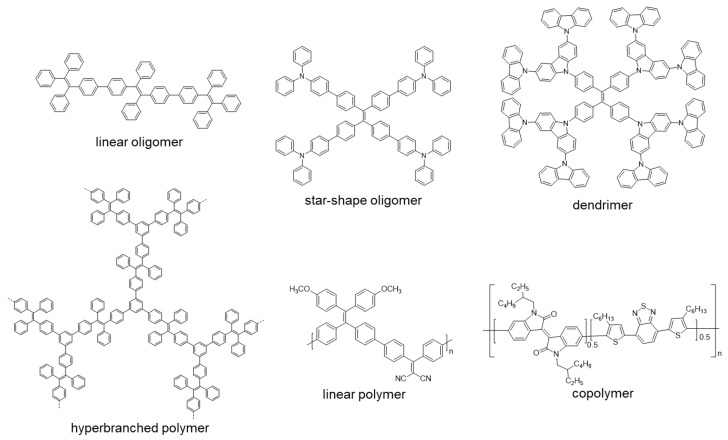
Different examples of linear and star-shape oligomers, dendrimers, linear and hyperbranched polymers and copolymers.

**Figure 2 polymers-13-00213-f002:**
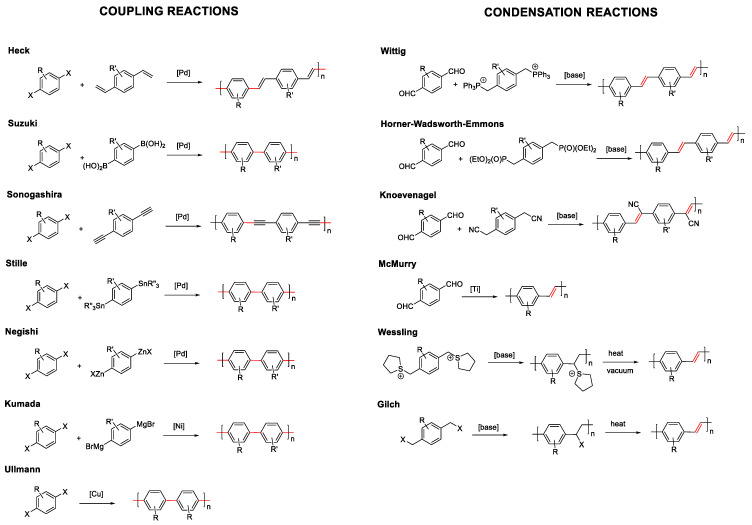
Synthetic methodologies for conjugated molecules. Coupling reaction on the left and condensation reaction on the right. Highlighted in red color the resulting formed bond.

**Figure 3 polymers-13-00213-f003:**
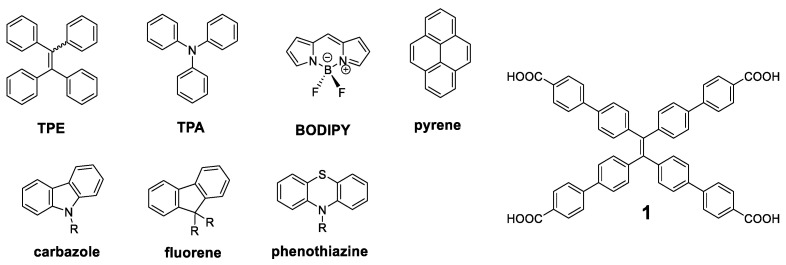
Structures of luminogens more widely used and of the compound **1**.

**Figure 4 polymers-13-00213-f004:**
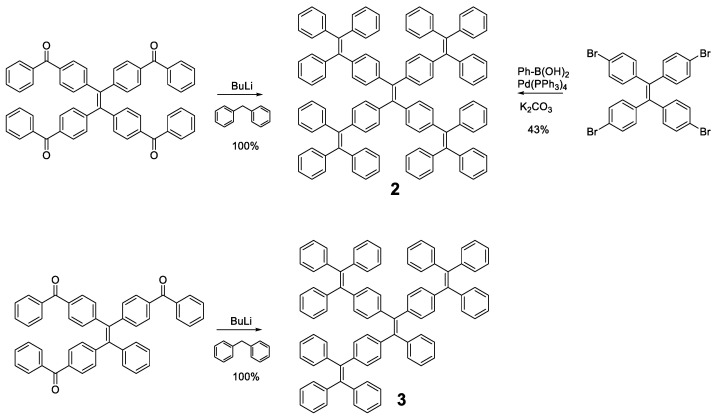
Preparation methods of **TPE**-cored oligomers **2** and **3**. On the left, the procedure described by Wang et al. [49] and on the right the one proposed by Tang et al. [50].

**Figure 5 polymers-13-00213-f005:**
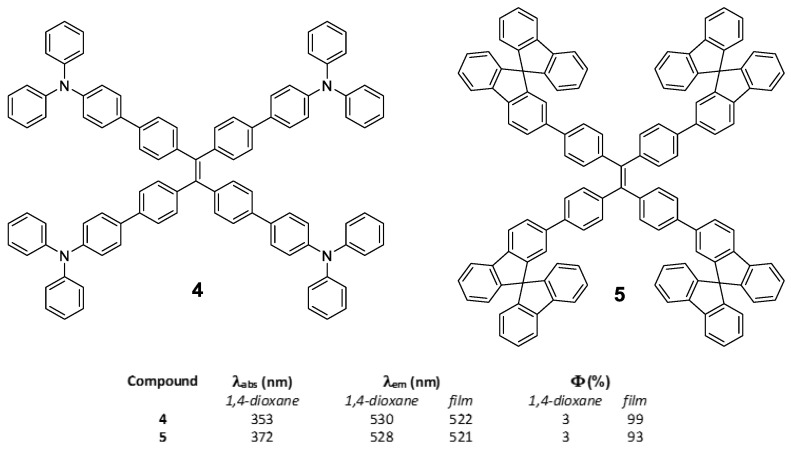
Structure of star shaped **TPE**-cored oligomers decorated with **TPA** (**4**) and 9,9′-spirobi[fluorene] (**5**) and table summarizing their photophysical properties in solution of 1,4-dioxane and in film.

**Figure 6 polymers-13-00213-f006:**
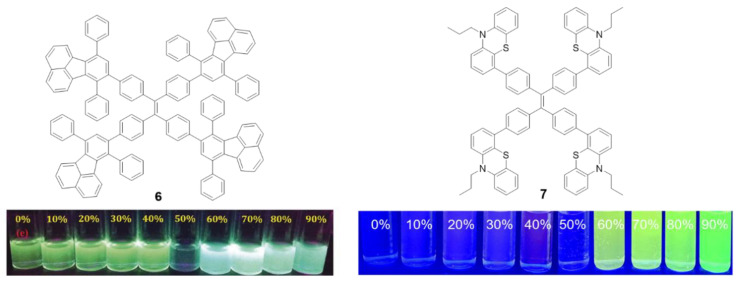
Structure of star shaped **TPE**-cored oligomers decorated with 7,10-diphenylfluoranthene (**6**) and **phenothiazine** (**7**) moieties and photographs showing changes in their emission behavior at different ratios of THF/H_2_O under illumination with 365 nm UV light [51,52]. Reproduced with permission. Copyright RSC and Wiley-VCH Velag GmbH & Co.

**Figure 7 polymers-13-00213-f007:**
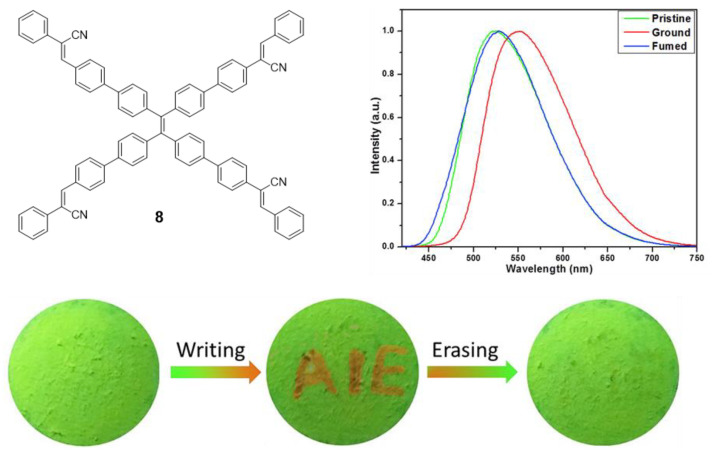
**TPE**-cored oligomer **8** decorated with four α-cyanostyrene units. Structure, photoluminescence spectra under different treatments: pristine, ground and exposed to CH_2_Cl_2_ vapors (fumed), and mechanofluorochromism: photographic images of fluorescence re-writable data recording on pieces of paper. Reproduced with permission from ref. [54]. Copyright Elsevier Ltd.

**Figure 8 polymers-13-00213-f008:**
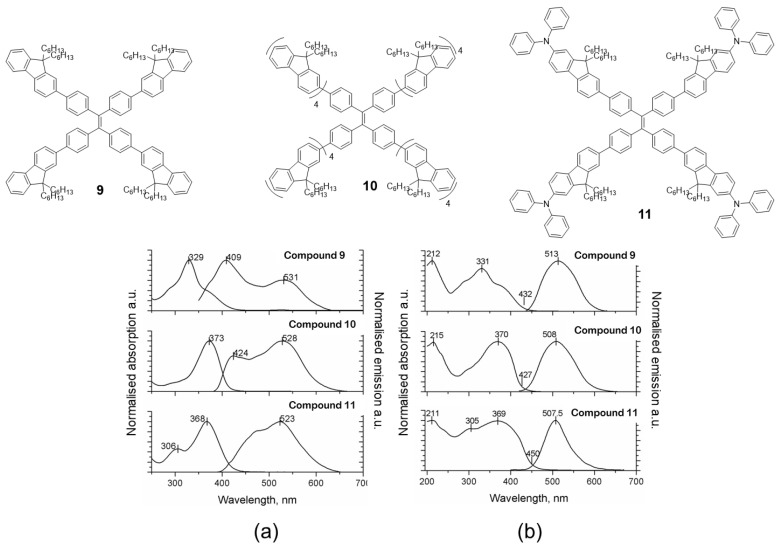
Examples of **TPE**-cored oligomers decorated with **fluorene** units. Compounds **9**, **10** and **11** and normalized absorption and photoluminescence spectra in dichloromethane (**a**) and solid state as films (**b**). Reproduced with permission from ref. [55]. Copyright Wiley-VCH Velag GmbH & Co.

**Figure 9 polymers-13-00213-f009:**
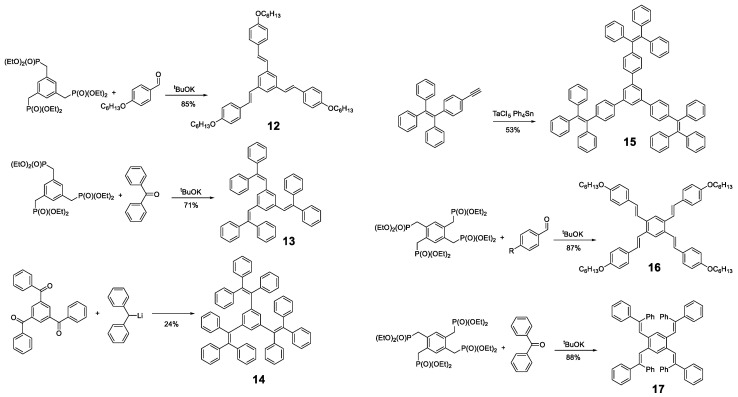
Examples of synthetic procedures to prepare benzene-cored oligomers involving catalyzed cyclotrimerization reaction (**15**), Horner-Wadsworth-Emmons reaction (**12**, **13**, **16** and **17**) and condensation with diphenylmethyl lithium (**14**).

**Figure 10 polymers-13-00213-f010:**
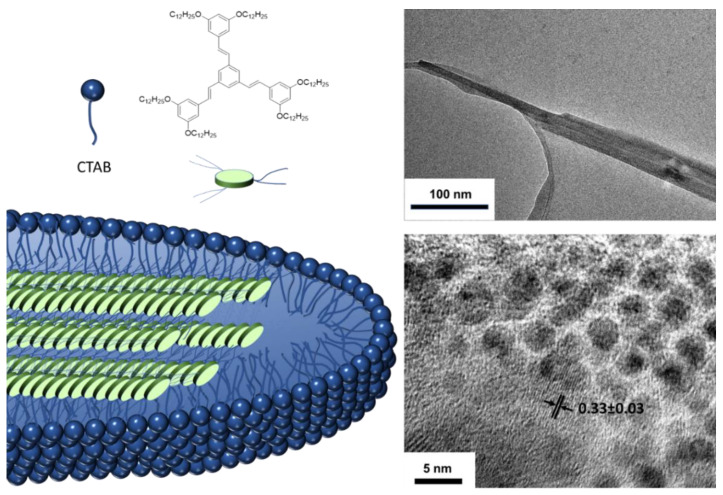
Aggregation induction through the formation of micelles and the use of CTAB as a surfactant. Photographs of TEM where the worm-shaped micelles are observed and the separations between the lattices formed. Reproduced with permission from [58]. Copyright Elsevier Ltd.

**Figure 11 polymers-13-00213-f011:**
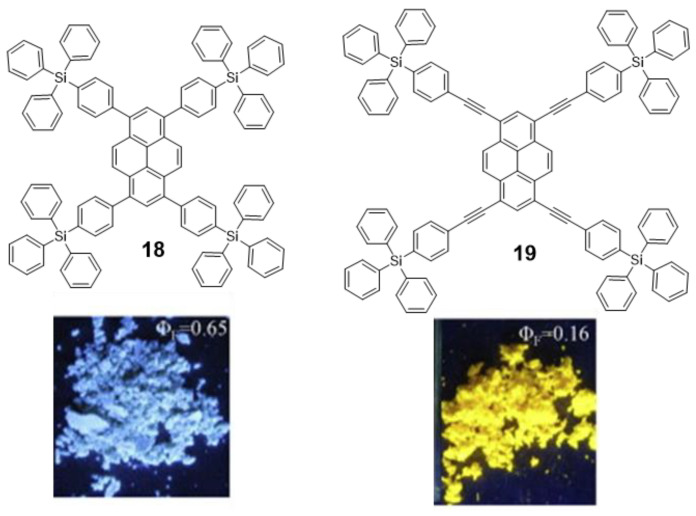
Structures of **pyrene**-cored oligomers **18** and **19** decorated with tetraphenylsilane units and a photograph of the solid powder of both compounds under UV light irradiation. Reproduced with permission from [64]. Copyright Elsevier Ltd.

**Figure 12 polymers-13-00213-f012:**
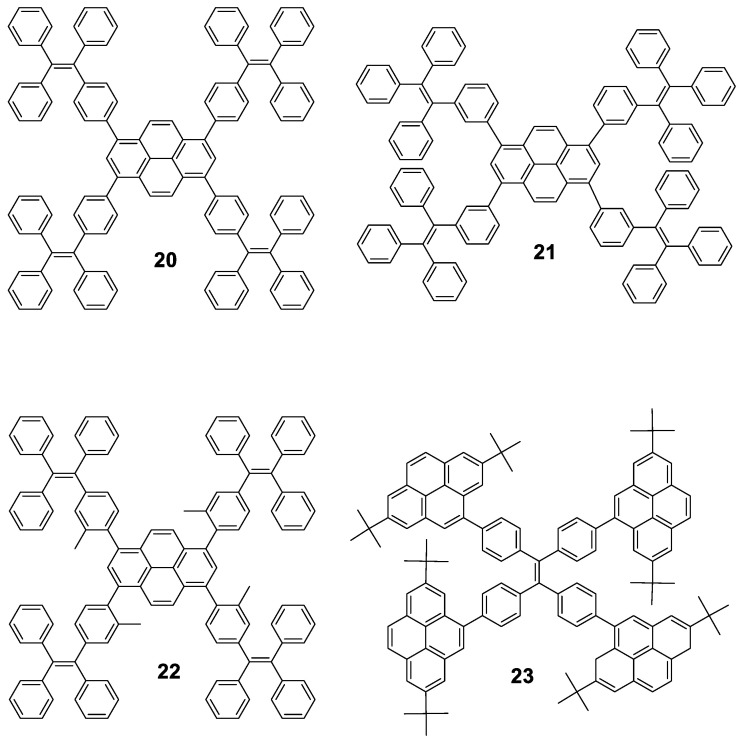
Combination of **pyrene** and **TPE** in the construction of fully conjugated molecules. Star shaped **pyrene**-cored oligomers **20**, **21** and **22** decorated with four **TPE** units, and **TPE**-cored oligomer **23** decorated with four **pyrene** units.

**Figure 13 polymers-13-00213-f013:**
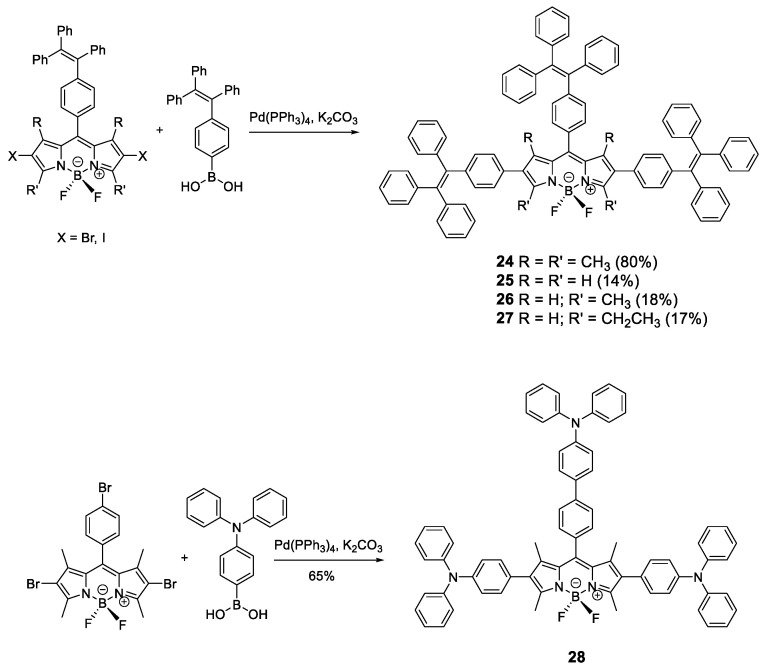
Synthetic scheme for **BODIPY** cored oligomers decorated with **TPE** (**24**–**27**) or **TPA** (**28**) luminogens.

**Figure 14 polymers-13-00213-f014:**
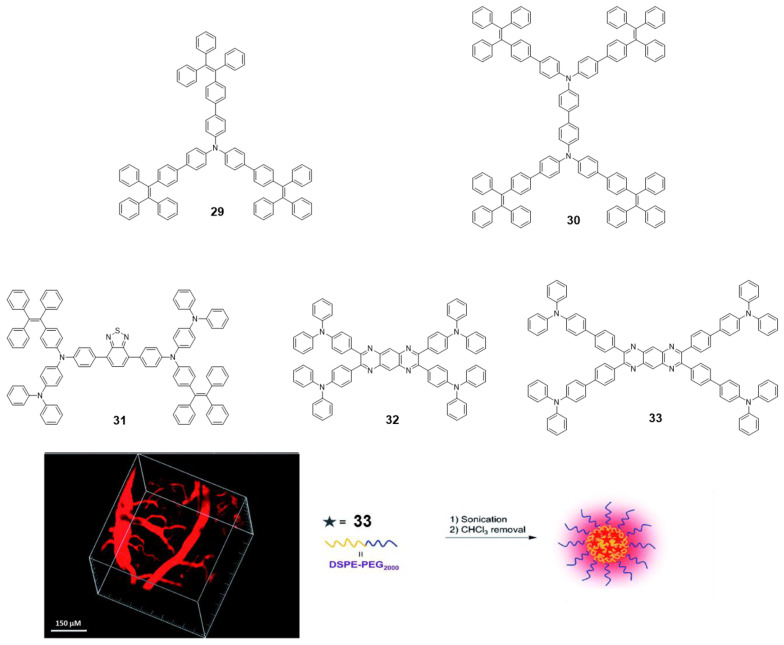
Schematic representation of **29**–**33**. On left, the constructed 3D image of the blood vessels of the mouse brain, 0.5 h after injection of the **31** dots. On right, schematic illustration of the fabrication of nanoparticles with **33** and DSPE-PEG2000. Reproduced with permission from ref. [74] and ref. [75]. Copyright the Royal Society of Chemistry.

**Figure 15 polymers-13-00213-f015:**
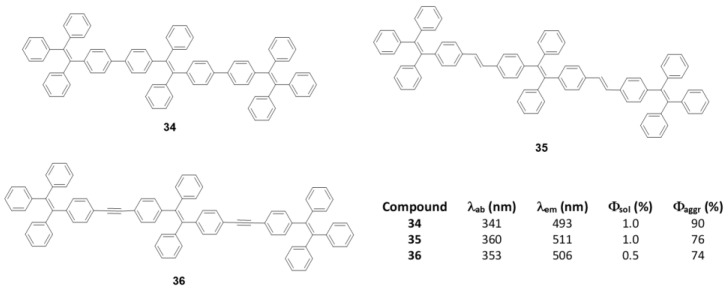
Linear **TPE**–oligomers **34**, **35**, and **36** and summary table of their spectroscopic data in THF and THF/water mixture *f_w_* = 90%.

**Figure 16 polymers-13-00213-f016:**
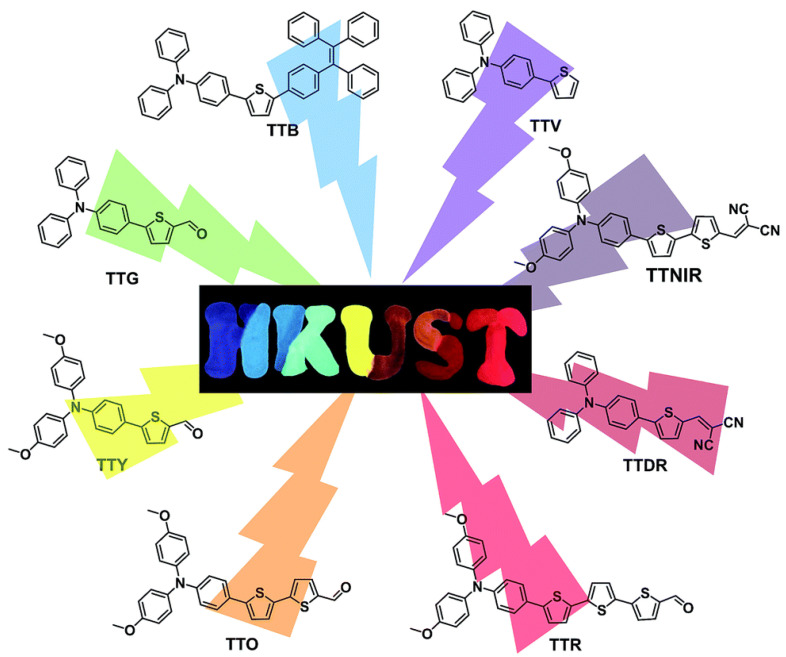
**TPA**–oligothiophenes with widely tunable emissions. Reproduced with permission from ref. [77]. Copyright the Royal Society of Chemistry.

**Figure 17 polymers-13-00213-f017:**
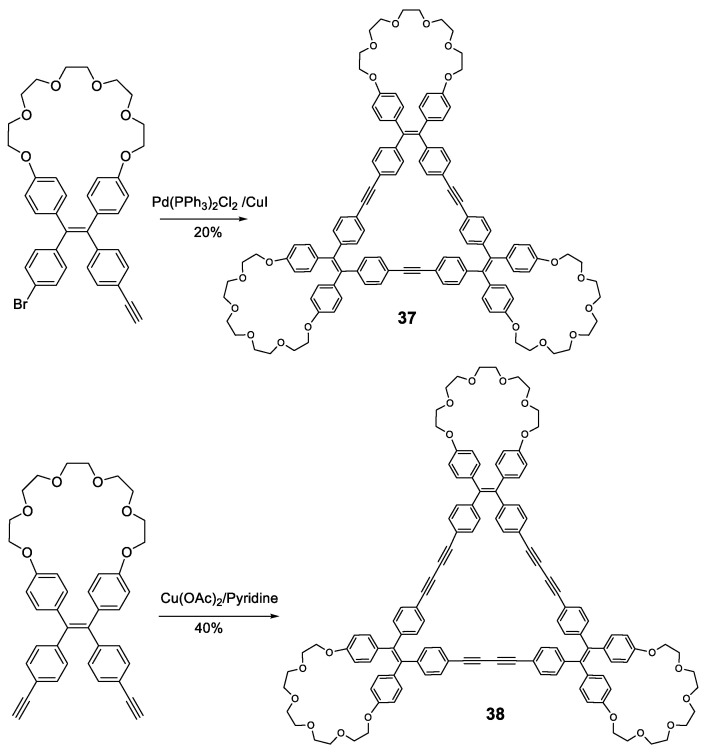
Synthetic scheme for **TPE**-crown ether macrocycles **37** and **38**.

**Figure 18 polymers-13-00213-f018:**
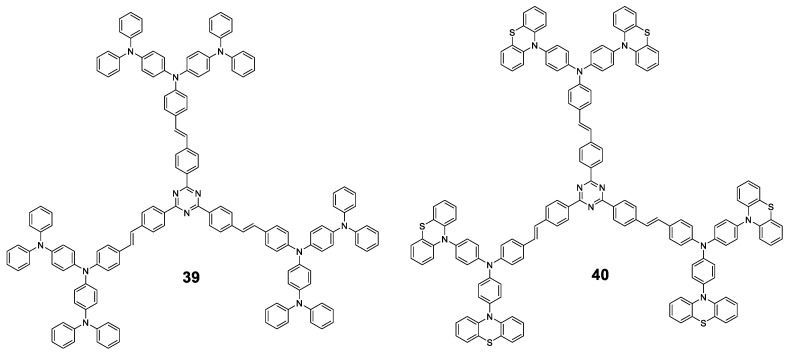
Dendrimeric structures **39** and **40** with **TPA** as the AIEgen moiety.

**Figure 19 polymers-13-00213-f019:**
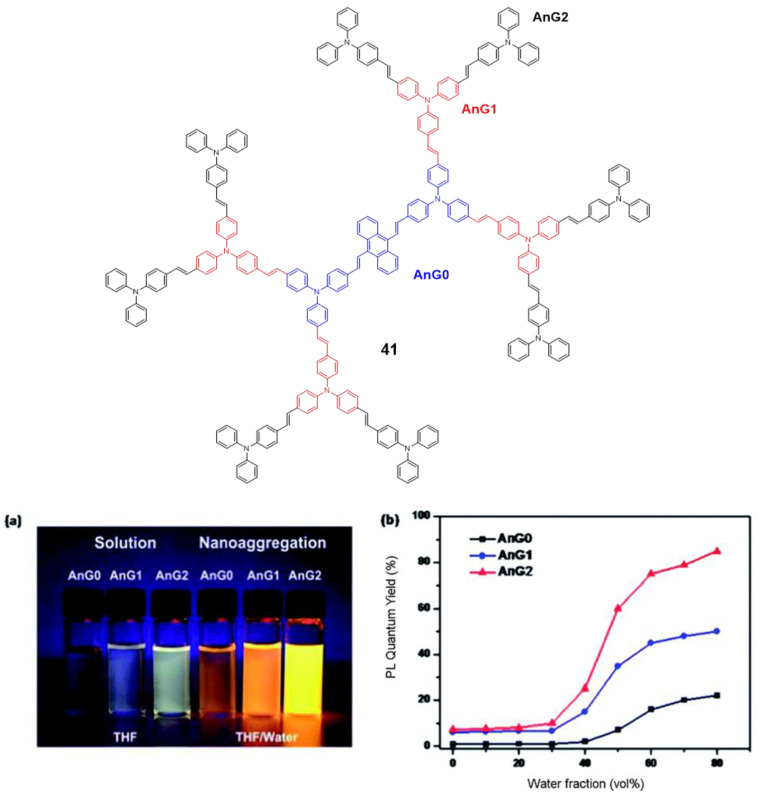
**TPA**–based dendrimer **41** highlighting the different generations derived from the linear core AnG0 (**a**) photograph of fluorescence emission in THF and THF/water mixture (80% water) under the UV light (365 nm) of the different dendrimer generations. (**b**) quantum yields as a function of the water fraction in THF for each dendrimer generation. Reproduced with permission from [91]. Copyright the Royal Society of Chemistry.

**Figure 20 polymers-13-00213-f020:**
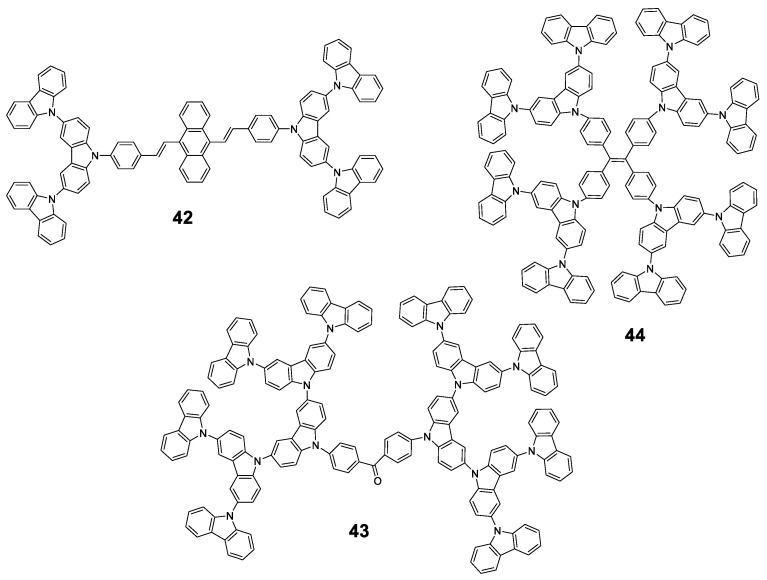
Dendrimers **42–44** with carbazole as AIEgen and different core moieties.

**Figure 21 polymers-13-00213-f021:**
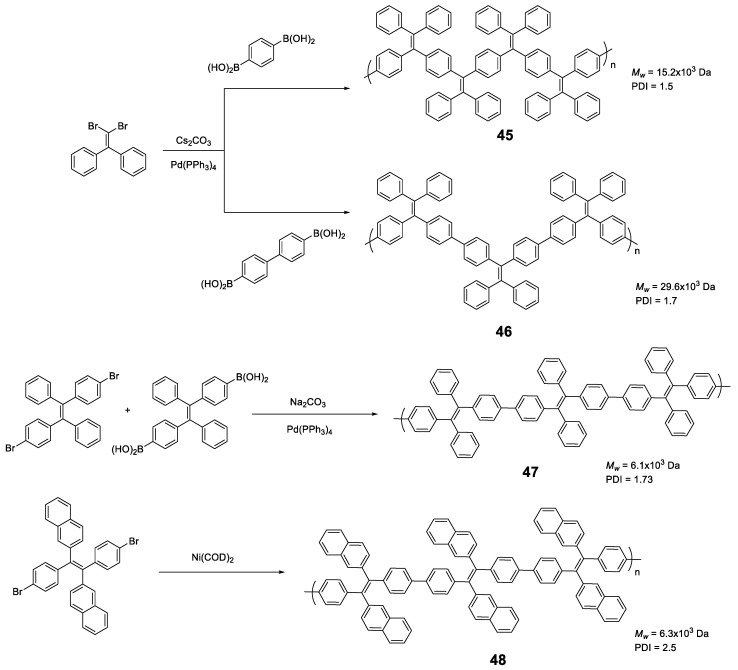
Synthetic scheme for the preparation of **TPE**-based polymers **45**–**48**.

**Figure 22 polymers-13-00213-f022:**
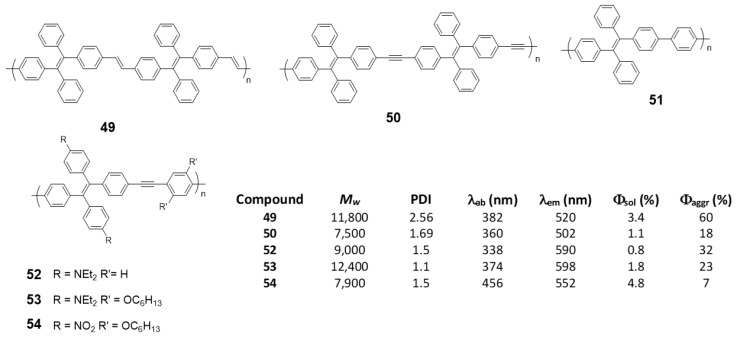
**TPE**-based linear polymers and summarized table with *M_w_*, PDI and spectroscopic data in THF solutions and THF/water mixtures *f_w_* = 99%.

**Figure 23 polymers-13-00213-f023:**
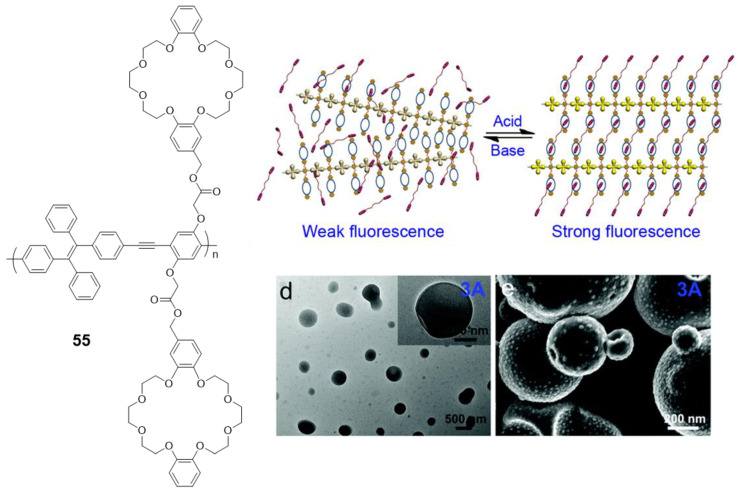
Crown ether-functionalized polymer **55** that self-assembles in a reversible way and photograph of the vesicles formed. Reproduced with permission from ref. [100]. Copyright the Royal Society of Chemistry.

**Figure 24 polymers-13-00213-f024:**
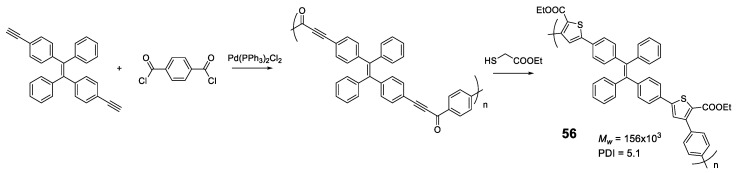
Scheme of one-pot acyl-Sonogashira cross-coupling and Fiesselmann cyclocondensation tandem reactions for the preparation of **56**.

**Figure 25 polymers-13-00213-f025:**
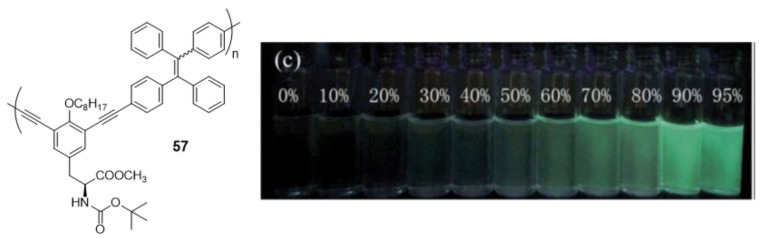
Compound **57** and photograph of its fluorescence in different THF/water mixtures under UV illumination (365 nm). Reproduced with permission from ref. [104]. Copyright the Royal Society of Chemistry.

**Figure 26 polymers-13-00213-f026:**
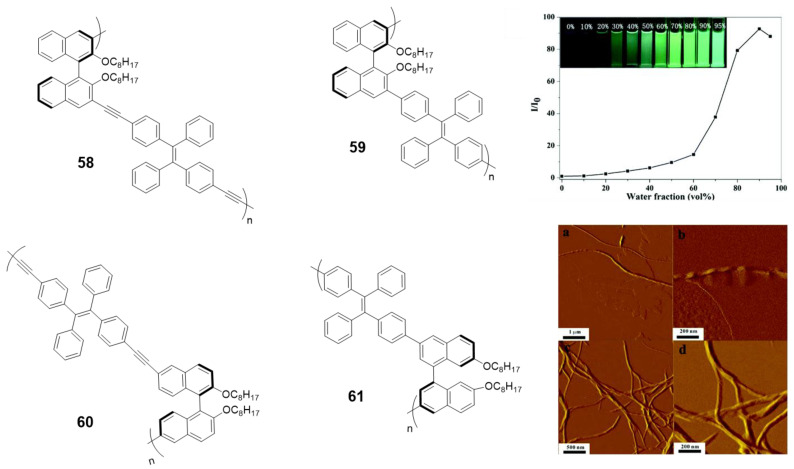
TPE-BINOL copolymers **58–61** and plot of (*I*/*I*_0_) values vs. water fractions (insert: photograph of fluorescence of **58** in different THF: water mixtures under UV illumination (365 nm)). AFM images of **58** obtained from THF/water: **a**, **b** in *f_w_* = 40 and **c**, **d** in *f_w_* = 60. Reproduced with permission from ref. [105]. Copyright the Royal Society of Chemistry.

**Figure 27 polymers-13-00213-f027:**
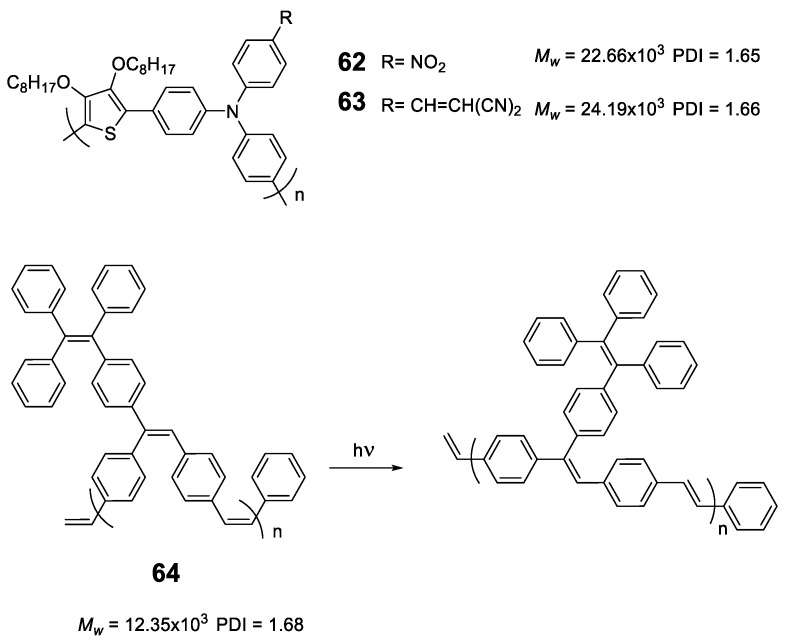
Structure of thiophene-TPA copolymers **62** and **63** and photoisomerization process for polyphenylene containing pending TPE groups in compound **64**.

**Figure 28 polymers-13-00213-f028:**
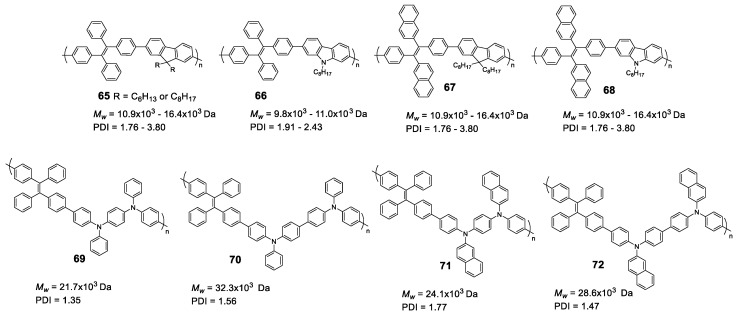
Structures of TPE in copolymers with other AIEgens such as dialkyl-fluorenes (**65** and **67**), *N*-alkyl carbazoles (**66** and **68**) and TPA derivatives (**69**–**72**).

**Figure 29 polymers-13-00213-f029:**
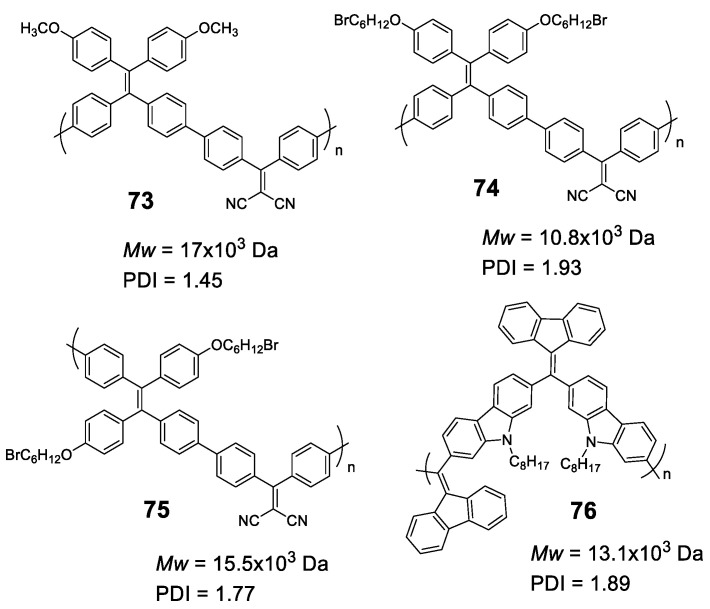
Structures of TPE-based copolymers showing electron donor-acceptor moieties in 73–75 and a modification of the TPE moiety through an *N*-alkyl carbazole group in **76**.

**Figure 30 polymers-13-00213-f030:**
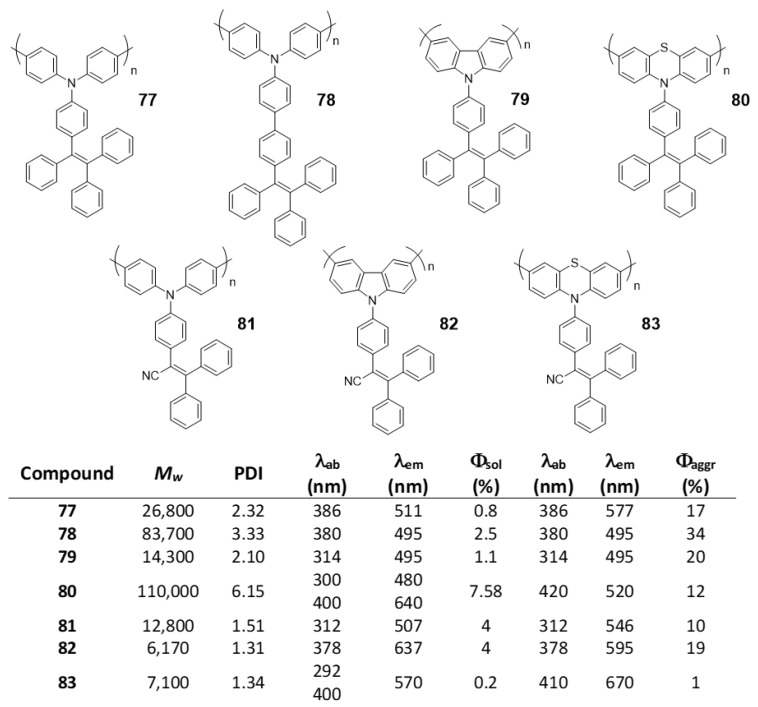
Compounds **77**–**83.** Summarizing table with *M_w_*, PDI and spectroscopic data in CHCl_3_ solutions and CHCl_3_/Ethanol mixtures for compounds **77** and **78** and THF solutions and THF/water mixtures for compounds **79**–**83**.

**Figure 31 polymers-13-00213-f031:**
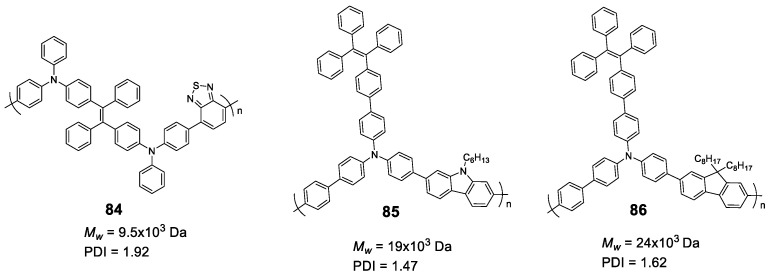
Structures of polymers including **TPA**, **TPE** and other luminogens such as dioctylfluorene (**84**), *N*-hexylcarbazole (**85**) and benzothiadiazole (**86**).

**Figure 32 polymers-13-00213-f032:**
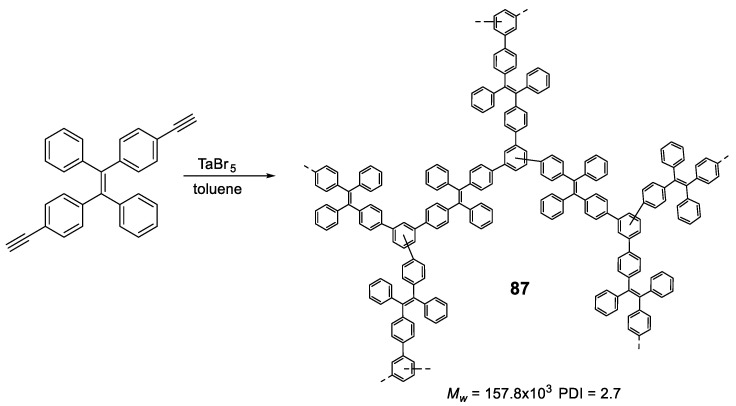
Synthetic scheme of polymer **87**.

**Figure 33 polymers-13-00213-f033:**
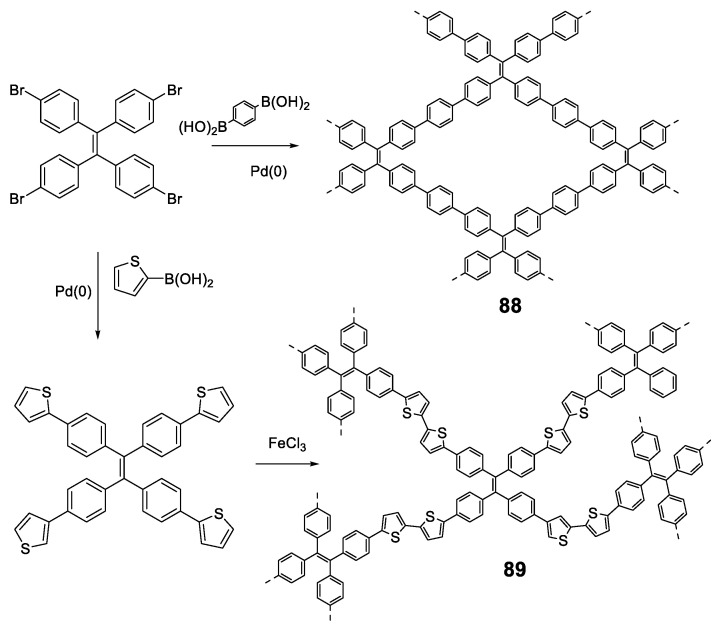
Synthetic routes for polymers **88** and **89** starting from common precursor tetrakis-(4-bromophenyl)ethylene.

**Figure 34 polymers-13-00213-f034:**
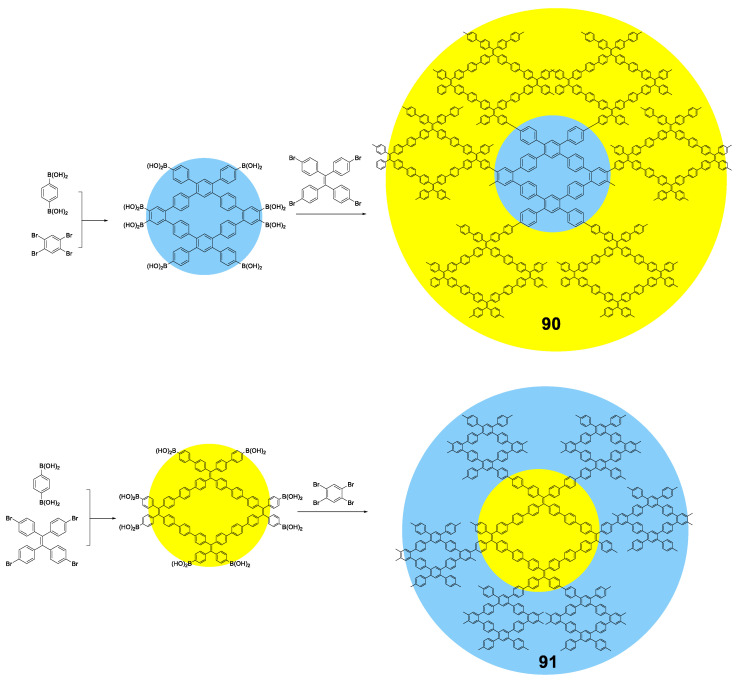
Synthetic procedures for producing core-shell nanoparticle **90** and its structurally-reversed **91**.

**Figure 35 polymers-13-00213-f035:**
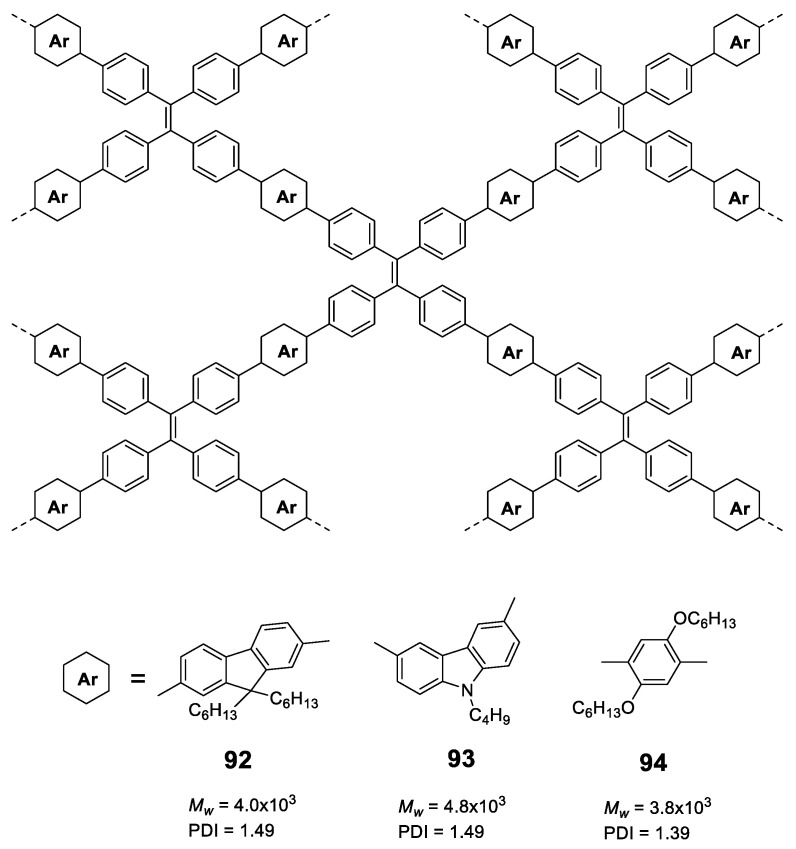
**TPE**-based polymers including additional AIEgens such as dihexylfluorene (**92**), *N*-butylcarbazole (**93**) and 2,5-dihexyloxyphenyl (**94**) groups.

**Figure 36 polymers-13-00213-f036:**
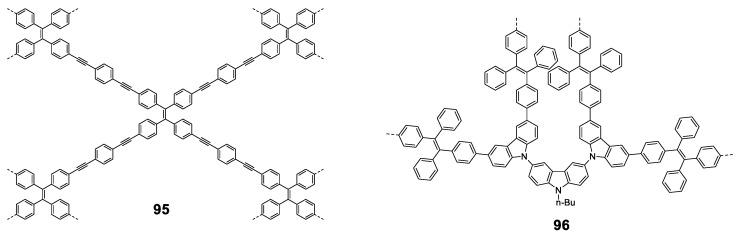
Polymeric materials based on TPE linked through 1,4-diethynylphenol (**95**) and copolymerized with carbazole derivatives in **96**.

**Figure 37 polymers-13-00213-f037:**
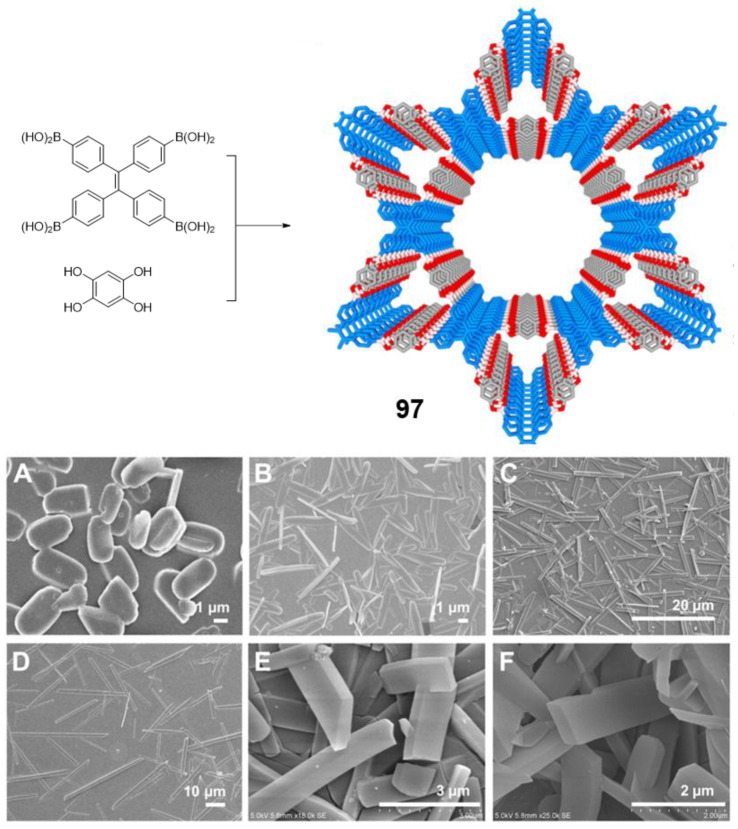
(**A**–**F**) Crystal structures and SEM images of AIE-active COF **97** built through the polymerization of **TPE**-boronic acid with 1,2,4,5-tetrahydroxyphenol. Reprint adapted with permission from ref. [138]. Copyright 2016 American Chemical Society.

**Figure 38 polymers-13-00213-f038:**
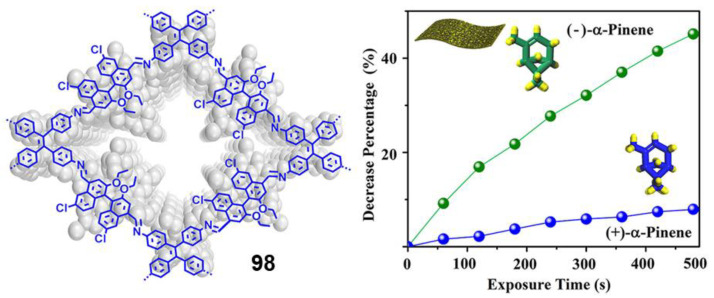
Views of the corresponding refined 2D crystal structures of **98** and decrease percentage of the fluorescence upon exposure to α-pinene for (R)-**98**. Reprint adapted with permission from ref [139]. Copyright 2019 American Chemical Society.

**Figure 39 polymers-13-00213-f039:**
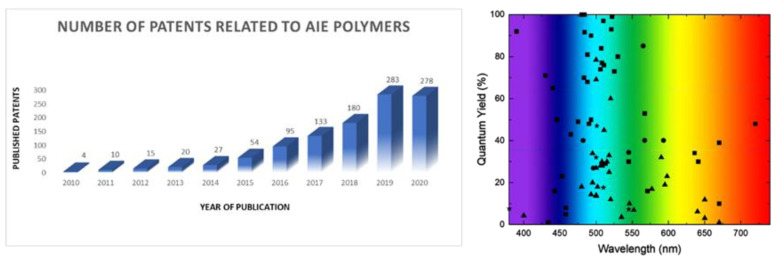
On the **left**, number of patents published in the last ten years with AIE polymers in the description (source: Google Patents). On the **right**, representation of quantum yield vs. maximum emission wavelength of the oligomers (square ■), dendrimers (circle ●), linear polymers (triangle ▲) and hyperbranched polymers (star ★) summarized in this review.

**Table 1 polymers-13-00213-t001:** Maximum absorption wavelength (λ_ab_), maximum emission wavelength (λ_em_), and fluorescence quantum yield (Φ_sol_) in THF solutions (1 μM) and maximum absorption wavelength (λ_ab_), maximum emission wavelength (λ_em_), and fluorescence quantum yield (Φ_f_) in aggregated THF/water mixtures, except for (a) that are in solid film.

Compound	λ_ab_ (nm)	λ_em_ (nm)	Φ_sol_ (%)	λ_exc_ (nm)	λ_em_ (nm)	Φ_F_ (%)
	THF Solution			Aggregated		
**12** ^1^	329	399 420	60	339 353	430	71
**13** ^2^	313	424	<1	348 ^a^	434 ^a^	<1 ^a^
**14** ^3^	310	450	0.09	330	480	100
**15** ^4^	326	488	0.26	-	-	81
**16** ^1^	350	461	43	440	461 467	43
**17** ^2^	340	467 488	<1	398^a^	488^a^	68 ^a^

^1^ ref. [15]. ^2^ ref. [59]. ^3^ ref. [60]. ^4^ ref. [61].

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
