# Peer review of "Aggregation-Induced Emission Properties in Fully π-Conjugated Polymers, Dendrimers, and Oligomers"

_polymers, 2021, doi:10.3390/polym13020213_

Round 1

Reviewer 1 Report

In this review by Sánchez-Ruiz and colleagues, the authors discuss how fully conjugated dendrimers, oligomers and polymers experience aggregation-induced emission. This is a very thoroughly written article, which is absolutely appropriate for this journal (Polymers). I am convinced that the readers will find it of considerable use as the topics considered in this work are very timely and interesting. I must say that I really liked the narration as it is easy to follow and very informative.

With all this in mind, I would like to offer some minor comments, which can improve this work further:
1) The authors have extensive knowledge on this topic, so it would be useful to discuss in more detail what is the potential for these chemicals for commercialization. Please inform the readers how close we are to the wide commercial implementation of these species. In fact, a SWOT analysis would be helpful to include. This would highlight all their pros and cons.
2) Since it is a review article, it would be good to make it a bit more critical. The above-mentioned SWOT analysis is a good opportunity to show what are the challenges that should be handled. If possible, the authors may also extend the main text and the references they describe by including the weak points of these articles.
3) Several pages have quite a lot of empty space. Please arrange the article in an appropriate way, so that it looks better when published.
4) Some of the plots extend beyond the margins e.g. Fig. 18 - please correct it.

Please do not treat the major revision in a negative way as it is a good paper overall. I just want to make sure that I can offer some further comments when the manuscript comes back from revision.

Author Response

We greatly appreciate the comments and feedback from referee #1 regarding our review. His suggestion concerning a SWOT analysis is very interesting. Although the research of this type of compounds is still in a very preliminary stage and there is not enough information to do a complete SWOT, we have incorporated a few early ideas about it. A section entitled Strengths, Weaknesses, Opportunities, and Threats in AIE polymers has been added. We really believe that the proposal of referee #1 has been very successful and that it improves the review a lot. We hope that it will be to the pleasure of the referee and the readers of the journal. Additionally, we have corrected the rest of the errors he points out. Regarding the out-of-margin graphics, it seems that it was an error in the conversion to the pdf document. We attach the manuscript with the additions highlighted in yellow.

Reviewer 2 Report

This concisely written review summarizes in a most comprehensive manner the current status of fully p-conjugated oligomers, dendrimers, and polymers that exhibited unique aggregation-induced emission properties. These p-conjugated large molecules with AIE properties have recently acquired prominence in the context of material science. Since this review is well covered in their synthetic routes, fluorescence properties, and potential applications, it will be certainly welcomed by researchers in the area of polymer chemistry and this special issue in Polymers. I suggest citing some related reviews (the authors mentioned at least 40 reviews have been published). In particular, quite recently, there have been published a few reviews for the similar p-conjugated large molecules with AIE properties (i.e. Chem. Asian J. 2019, 14, 715, Polym. Chem. 2019, 10, 3822, Coord. Chem. Rev. 2020, 402, 213076, Prog. in Polym. Sci. 2020, 100, 101176, Chinese J. Polym. Sci. 2019, 37, 289, ChemistrySelect 2019, 4, 12848.).

Minor points:

  • Please add the product yields for the related compounds in Figures 4, 9, 13, and 17.
  • Please show the solvent in the measurement of quantum yields in Figures 15, 22, and 30.
  • “Mw” should be the italic form in Figure 21.
  • A hyphen should be corrected in Figure 35.

Author Response

We appreciate the comments of referee #2 and have modified the manuscript to reflect his suggestions. The suggested references have been included, as well as the reaction yields and the solvents used in the figures mentioned. Finally, we have corrected the typographical errors. All the changes incorporated in the manuscript have been highlighted in yellow. Please see the attachment. Finally, I would like to wish you a great 2021.

Reviewer 3 Report

The manuscript provides a very comprehensive overview of conjugated oligomers, dendrimers or other small molecules, and polymers, with interesting fluorescence properties, focusing in aggregation-induced luminescence The manuscript also includes a section about synthetic methods reviewing the general strategies to prepare star-shaped small molecules and dendrimer-like polymers with AI-E properties. The manuscript is very well written, and the presentation of the materials is under a logical manner, also including historical notes about their discoveries, which increases the interest of the manuscript. The relationships between the molecular design and luminescence properties are well explained for an article more focused on the materials ‘structural characteristics. My opinion is that the manuscript deserves publication and I only have a minor comment. This refers to the captions of the figures presenting the chemical structures. On such, the authors chose very brief comments that cannot stand alone without the related text (e.g. . “Compounds 18 and 19 and a photograph of … “.) My suggestion is to modify them in order to make them more informative and therefore give more relevance to the figures (e.g. examples of star-shaped small molecules with pyrene central units exhibiting …. ) .

Author Response

We appreciate the comments of referee #3 and have taken up his suggestion to improve the manuscript. We have improved the footnotes in figures 4, 5, 6, 7, 8, 9, 11,12,13, 28, 29, 31, 32, 33, 34, 36, and 37. The added text has been highlighted in yellow in the manuscript. Please see the attachment. Finally, I would like to wish you a great 2021.

Round 2

Reviewer 1 Report

Thank you for following the suggestions. It is an excellent contribution, so I would like to recommend its publication in Polymers.